



# Airborne glyoxal measurements in the marine and continental atmosphere: Comparison with TROPOMI observations and EMAC simulations

Flora Kluge[1], Tilman Hüneke[1,a], Christophe Lerot[2], Simon Rosanka[3], Meike K. Rotermund[1], Domenico Taraborrelli[3], Benjamin Weyland[1], and Klaus Pfeilsticker[1,4]

[1]Institute of Environmental Physics, Heidelberg University, Heidelberg, Germany
[2]Atmospheric reactive gases, Royal Belgian Institute for Space Aeronomy (BIRA-IASB), Brussels, Belgium
[3]Forschungszentrum Jülich, Institute for Energy and Climate Research: Troposphere (IEK-8), Jülich, Germany
[4]Heidelberg Center for the Environment, Heidelberg University, Heidelberg, Germany
[a]now with Encavis AG, Hamburg, Germany

**Correspondence:** Flora Kluge
(fkluge@iup.uni-heidelberg.de)

**Abstract.** We report on airborne Limb and Nadir measurements of vertical profiles and total vertical column densities (VCDs) of glyoxal ($C_2H_2O_2$) in the troposphere, which were performed from aboard the German research aircraft HALO (High Altitude and LOng Range) in different regions and seasons around the globe between 2014 and 2019. The airborne Nadir and integrated Limb profiles excellently agree among each other. Our airborne observations are further compared to collocated gly-

oxal measurements of the TROPOspheric Monitoring Instrument (TROPOMI), with good agreement between both data sets for glyoxal observations in (1) pristine terrestrial, (2) pristine marine, (3) mixed polluted, and (4) biomass burning affected air masses with high glyoxal concentrations. Exceptions from the overall good agreement are observations of (1) faint and aged biomass burning plumes over the oceans and (2) of low lying biomass burning or anthropogenic plumes in the terrestrial or marine boundary layer, both of which contain elevated glyoxal that is mostly not captured by TROPOMI. These differences

of airborne and satellite detected glyoxal are most likely caused by the overall small contribution of plumes of limited extent to the total atmospheric absorption by glyoxal and the difficulty to remotely detect weak absorbers located close to low reflective surfaces (e.g. the ocean in the visible wavelength range), or within dense aerosol layers. Observations of glyoxal in aged biomass burning plumes (e.g. observed over the Tropical Atlantic off the coast of West Africa in summer 2018, off the coast of Brazil by the end of the dry season 2019, and the East China Sea in spring 2018) could be traced back to related

wildfires, such as a plume crossing over the Drake Passage that originated from the Australian bushfires in late 2019. Our observations of glyoxal in these over days aged biomass burning plumes thus confirm recent findings of enhanced glyoxal and presumably secondary aerosol (SOA) formation in aged wildfire plumes from yet to be identified longer-lived organic precursor molecules (e.g. aromatics, acetylene, or aliphatic compounds) co-emitted in the fires. Further, elevated glyoxal (median 44 ppt) as compared to other marine regions (median 10–19 ppt) is observed in the boundary layer over the tropical oceans,

well in agreement with previous reports. The airborne data sets are further compared to glyoxal simulations performed with the global atmosphere-chemistry model EMAC (ECHAM/MESSy Atmospheric Chemistry). When using an EMAC setup that





resembles recent EMAC studies focusing on complex chemistry, reasonable agreement is found for pristine air masses (e.g. the unperturbed free and upper troposphere), but notable differences exist for regions with high emissions of glyoxal and glyoxal producing volatile organic compounds (VOC) from the biosphere (e.g. the Amazon), mixed emissions from anthropogenic activities (e.g. over continental Europe, the Mediterranean and East China Sea), and potentially from the sea (e.g. the tropical oceans). Also, the model tends to largely under-predict glyoxal in city plumes and aged biomass burning plumes. The potential causes for these differences are likely to be multifaceted, but they all point to missing glyoxal sources from the degradation of the cocktail of (potentially longer-chained) organic compounds emitted from anthropogenic activities, biomass burning, and from the organic micro-layer of the sea.

## 1 Introduction

Glyoxal ($C_2H_2O_2$), the simplest $\alpha$-dicarbonyl, has significant importance in air quality and climate, due to its role as an intermediate in the oxidation of hydrocarbons (e.g. Finlayson-Pitts and J. N. Pitts (1986); Volkamer et al. (2001); Fu et al. (2008); Myriokefalitakis et al. (2008); Vrekoussis et al. (2009); Nishino et al. (2010); Li et al. (2016); Chan Miller et al. (2017); Wennberg et al. (2018)) and as an important precursor for secondary organic aerosol (SOA) formation and thus for the aerosol forcing of climate (e.g. Jang and Kamens (2001); Liggio et al. (2005b); Volkamer et al. (2007); Lim et al. (2013); Knote et al. (2014); Kim et al. (2022)).

The global sources of glyoxal are estimated to $45\,\mathrm{Tg\,a^{-1}}$ (Fu et al., 2008), and the largest single source ($\sim 47\%$) is believed to be the oxidation of isoprene emitted by vegetation (e.g. Myriokefalitakis et al. (2008); Li et al. (2016); Chan Miller et al. (2017); Wennberg et al. (2018)). Precursor molecules of glyoxal, that are mostly (but not exclusively) anthropogenically emitted include alkenes, acetylene, various aromatics, monoterpenes and other volatile organic compounds (VOCs) with different yields (Volkamer et al., 2001; Fu et al., 2008; Nishino et al., 2010; Taraborrelli et al., 2021). A recent study found that below $2\,\mathrm{km}$ altitude, the production of glyoxal in the city plume of the Seoul Metropolitan Area (South Korea) was mainly caused by the oxidation of aromatics ($\sim 59\%$) initiated by hydroxyl radicals (Kim et al., 2022). Glyoxal is also directly emitted in considerable amounts by biomass burning, together with a suite of organic glyoxal precursor molecules in largely seasonal and regional varying amounts (e.g. Andreae (2019); Akagi et al. (2011); Stockwell et al. (2015); Zarzana et al. (2017, 2018); Kluge et al. (2020)).

The predominant photochemical loss process of glyoxal is photolysis, and to a lesser degree reactions with OH radicals (Koch and Moortgat, 1998; Volkamer et al., 2005a; Tadić et al., 2006; Fu et al., 2008; Wennberg et al., 2018). Uptake of glyoxal on aerosols in polluted environments as well as on cloud particles can eventually compete with its photochemical losses (e.g. Volkamer et al. (2007); Kim et al. (2022)), primarily due to its high water solubility (e.g. Zhou and Mopper (1990); Kroll et al. (2005); Ip et al. (2009); Kampf et al. (2013)), and oligomerization potential (e.g. Whipple (1970); Liggio et al. (2005a); Loeffler et al. (2006); Galloway et al. (2009)). While the global mean lifetime of glyoxal is less than a few hours, in the sunlit polluted atmosphere it can be as short as half an hour due to photolysis, reactions with OH and heterogeneous uptake (Volkamer et al., 2007; Kim et al., 2022).





Accordingly, due to the varying source strength of glyoxal and its short lifetime, in pristine air its mixing ratios may range from several ppt to a few 10 ppt. For example, 7–23 ppt of glyoxal were found over the South Pacific (Lawson et al., 2015), or up to 10 ppt at the Cape Verde Atmospheric Observatory (CVAO, Sao Vicente island) over the Tropical Atlantic (Walker et al., 2022). Further, Mahajan et al. (2014) reported an average glyoxal mixing ratio of 25 ppt from 10 field campaigns over the open oceans in different parts of the world. Contrary to the low glyoxal mixing ratios observed in the pristine marine environment,

in polluted air glyoxal mixing ratios may reach several 100 ppt (e.g. Lee et al. (1998); Volkamer et al. (2005a, 2007); Fu et al. (2008); Sinreich et al. (2010); Baidar et al. (2013); Kaiser et al. (2015); Volkamer et al. (2015); Chan Miller et al. (2017); Kluge et al. (2020); Kim et al. (2022), and others), or even up to 1.6 ppb, as found over a tropical rainforest with large emissions of isoprene in South-East Asia (MacDonald et al., 2012).

    Glyoxal is detectable from space by satellites applying a similar technique (DOAS, Differential Optical Absorption Spec-

troscopy, (Platt and Stutz, 2008)) as used for the airborne data in this study. Accordingly, since the first glyoxal observations of UV/vis Nadir observing spectrometers (e.g. from the Scanning Imaging Absorption Spectrometer for Atmospheric Chartography (SCIAMACHY); Wittrock et al. (2006)), numerous studies with ever increasing observation capabilities and spatial resolution have been reported for space-borne measurements of vertical column densities (VCD) of glyoxal (e.g. from the instruments OMI (Ozone Monitoring Instrument), and TROPOMI (TROPOspheric Monitoring Instrument; Stavrakou et al.

(2009a); Vrekoussis et al. (2009); Lerot et al. (2010); Vrekoussis et al. (2010); Chan Miller et al. (2014); Alvarado et al. (2020, 2014); Lerot et al. (2021)), and many others). These space-borne measurements provided a wealth of new information on the worldwide sources, occurrence and abundance of glyoxal and its relation to the photochemistry of VOCs and aerosols (e.g. Fu et al. (2008)).

    Simultaneous measurements of formaldehyde ($CH_2O$) and glyoxal by satellites and airborne instruments complemented by

modelling have been exploited to study its different sources (e.g. Stavrakou et al. (2009b); Lerot et al. (2010); Boeke et al. (2011); Chan Miller et al. (2014); Bauwens et al. (2016); Stavrakou et al. (2016)), to elucidate the secondary aerosol formation from carbonyls (typically in the background atmosphere) as well as its fate in biomass burning plumes (Knote et al., 2014; Li et al., 2016; Lim et al., 2019), and even more recently, to estimate the organic aerosol abundance (Liao et al., 2019). Ground-based, air-, and space-borne simultaneous measurements of formaldehyde and glyoxal and of the $[C_2H_2O_2]/[CH_2O]$

ratio were further helpful to study the glyoxal yield from isoprene oxidation in relation to formaldehyde, to specify the various hydrocarbon glyoxal precursors, to investigate the anthropogenic impact on rural photochemistry, and more recently to investigate the sources of glyoxal and the chemical evolution of VOCs in biomass burning plumes (e.g. Lee et al. (1998); Wittrock et al. (2006); Vrekoussis et al. (2010); DiGangi et al. (2012); MacDonald et al. (2012); Kaiser et al. (2015); Bauwens et al. (2016); Li et al. (2016); Chan Miller et al. (2017); Zarzana et al. (2017); Hoque et al. (2018); Behrens et al. (2019); Alvarado

et al. (2020); Kluge et al. (2020)).

    Together with respective photochemical model simulations, some of the glyoxal observations point to deficits in our present understanding of atmospheric glyoxal. This includes (1) the observation of unexpected large glyoxal concentrations in the marine boundary layer of the Eastern Pacific up to 3000 km from the continental coast by ship-borne (up to 140 ppt; Sinreich et al. (2010)) and airborne measurements (32 – 36 ppt; Volkamer et al. (2015)). These measurements revealed a yet unknown ma-





rine source of glyoxal, possibly from ozone-driven reactions with the organic microlayer at the sea surface, which in idealised seawater laboratory experiments have been shown to produce glyoxal (Zhou et al., 2014), and/or secondary formation from oxidised VOC (OVOC) precursor molecules, such as acetaldehyde, acetylene, and others (Wang et al., 2019). (2) Unlike for formaldehyde, the models had still varying success in reproducing the glyoxal VCDs (Myriokefalitakis et al., 2008; Stavrakou et al., 2009a; Lerot et al., 2010). Moreover, in comparisons of satellite measurements from SCIAMACHY and GOME-2,

several studies have found that the models underestimate global glyoxal emissions, when not considering additional biogenic sources (Myriokefalitakis et al., 2008; Stavrakou et al., 2009a; Lerot et al., 2010). (3) Finally, a recent study by Alvarado et al. (2020) found unexpected large amounts of glyoxal (and formaldehyde) in several days old air masses originating from Canadian wildfires in August 2018, which can only be reconciled with the source strength and lifetime of glyoxal if considering secondary formation of glyoxal from OVOCs that were co-emitted from the fires.

Here, we report on airborne measurements of glyoxal concentrations (Limb) and total vertical column densities (Nadir) performed during eight missions from the German research aircraft HALO (High Altitude and Long Range) in different regions of the globe between 2014 and 2019. The measurements are able to provide novel information on the sources and fate of glyoxal in the atmosphere and address aspects on some open issues in glyoxal and VOC research as outlined above. This includes novel insights into the amount and vertical distribution of glyoxal in the polluted terrestrial (South America, Europe, East Asia),

polluted marine (East China and Mediterranean Sea, South American and West African coastlines), pristine terrestrial (South America), and pristine marine (South and North Atlantic) atmosphere. The observations may thus not only serve to validate respective satellite observations, but also simulations of atmosphere-chemistry models to better assess the global budget of glyoxal, the role of glyoxal for the atmospheric oxidation capacity and its contribution to secondary aerosol formation. Finally, our glyoxal profiles and VCDs measured in different regions and seasons may also serve as input for air mass calculations

necessary to better infer total atmospheric column densities of glyoxal from the various satellite observations (e.g. GOME-2, SCIAMACHY, OMI, and TROPOMI).

In the present study, the airborne Nadir glyoxal measurements are corroborated and validated against near collocated observations of the TROPOMI satellite instrument. The Limb and Nadir measurements are further compared to simulations of the global ECHAM/MESSy Atmospheric Chemistry (EMAC) model, of which further details are presented in a companion paper

(S. Rosanka et al., in preparation). The specific scientific questions addressed in both publications include: (1) the marine and terrestrial background of glyoxal and its potential sources, (2) the sources of glyoxal in the polluted atmosphere, (3) secondarily formed glyoxal in biomass burning plumes, (4) the global budget of glyoxal, and (5) its potential contribution to secondary aerosol formation.

The paper is organized as follows. Section 2 briefly describes the measurement technique and involved methods used in the

present study. Section 3 reports on the deployment and measurements of the mini-DOAS instrument on eight missions of the HALO research aircraft into different regions from 2014 until 2019. Chapter 4 presents the airborne concentration and VCD measurements of glyoxal. The latter are then compared to collocated total atmospheric column density observations of glyoxal made from the TROPOMI instrument. Finally, both airborne data sets -glyoxal concentrations and VCDs- are compared to





simulations of the global atmosphere-chemistry model EMAC. Chapter 5 discusses the major findings and results, and chapter
6 concludes and summarises the study.

## 2 Instruments and methods

### 2.1 The airborne mini-DOAS measurements

The airborne mini-DOAS measurements of UV/vis/near-IR absorbing gases involve: (1) simultaneous measurements of Limb
and Nadir scattered skylight, (2) the Differential Optical Absorption Spectroscopy (DOAS) analysis of the measured skylight
spectra for the target gases (Platt and Stutz, 2008), and (3) forward radiative transfer modelling of the observations using the
Monte Carlo model McArtim (Deutschmann et al., 2011). In a last step, trace gas concentrations are inferred from the Limb
observations by scaling the measured slant column densities (SCDs) using simultaneously measured SCDs of a scaling gas of
known concentration (e.g. $O_3$), or calculated (clear sky) extinction of the collisional complex $O_2-O_2$, or briefly ($O_4$; Hüneke
(2016); Hüneke et al. (2017); Stutz et al. (2017); Werner et al. (2017); Kluge et al. (2020); Rotermund et al. (2021)). For the
Nadir observations, air mass factors are simulated using the same radiative transfer model (McArtim) to infer VCDs from the
measured SCDs.

The presented study focuses on the Nadir and Limb measurements of $O_4$ and $C_2H_2O_2$ by the mini-DOAS instrument made
from aboard the German research aircraft HALO during a total of 72 research flights on eight scientific missions covering
different regions of the globe. The processing of the measured data is mainly based on our previous study on $O_4$, $CH_2O$, and
$C_2H_2O_2$ (Kluge et al., 2020). Since some aspects of the data processing changed since then, necessary details on these changes
and refinements are provided in the following.

### 2.1.1 The mini-DOAS instrument

The mini-DOAS instrument is a UV/vis/near-IR six channel optical spectrometer, which has been operated on board the
HALO research aircraft since 2011. It detects and spectrally analyses Nadir and Limb scattered sunlight in the UV (310–
440 nm, FWHM = 0.47 nm), visible (420–640 nm, FWHM = 1.1 nm) and near-infrared (1100–1680 nm, FWHM = 10 nm)
wavelength ranges (Hüneke, 2016; Hüneke et al., 2017). The six telescopes (FOV $0.5° \times 3.15°$) collect the skylight from
fixed Nadir and Limb viewing geometries each in the UV, visible and near-IR channel. The limb telescopes can be adjusted
to varying elevation angles (+5 to -90°) when commanded, but are normally aligned with a rate of 10 Hz at -0.5° below the
horizon to compensate for the changing roll angle of the aircraft. Glass fibre bundles transmit the collected skylight to six optical
spectrometers, assembled in an evacuated ($10^{-5}$ mbar) and thermostated housing (at $\sim 1°C$) in the otherwise umpressurized
and uninsulated boiler room of the aircraft. In the Limb geometry, the mini-DOAS instrument probes air-masses perpendicular
to the aircraft's flight direction on the starboard side, with typical photon path lengths in the visible wavelength range ranging
from $\sim 10$ km (at 2 km altitude) to about $\sim 100$ km at the maximum flight altitude of the aircraft around 15 km, depending on
the wavelength, aerosol concentration and cloud cover (see Fig. 2 in Kluge et al. (2020) and below). In the Nadir observation



**Table 1.** Trace gas absorption cross sections used for the spectral retrieval. For $H_2O$, gas phase (g) as well as liquid phase (l) absorption cross sections are used.

| No. | Absorber | Temperature [K] | Reference | Uncertainty [%] |
|---|---|---|---|---|
| 1a | $O_4$ | 273 | Thalman and Volkamer (2013) | 4 |
| 1b | $O_4$ | 293 | Thalman and Volkamer (2013) | 4 |
| 2a | $O_3$ | 203 | Serdyuchenko et al. (2014) | 3 |
| 2b | $O_3$ | 223 | Serdyuchenko et al. (2014) | 3 |
| 2c | $O_3$ | 273 | Serdyuchenko et al. (2014) | 3 |
| 3 | $NO_2$ | 294 | Vandaele et al. (1998) | 3 |
| 4a | $H_2O$ (g) | 293 | Rothman et al. (2009) | 8 |
| 4b | $H_2O$ (g) | 296 | Polyansky et al. (2018) | 1 |
| 4c | $H_2O$ (l) | 295 | Pope and Fry (1997) | 1 |
| 5 | $CH_2O$ | 293 | Chance and Orphal (2011) | 10 |
| 6 | $C_2H_2O_2$ | 296 | Volkamer et al. (2005b) | 3 |

**Table 2.** Details of the DOAS spectral analysis for the various trace gases.

| Spectrometer | Target gas | Spectral interval [nm] | Fitted absorbers (see Table 1) | Polyn. | Mission (i) |
|---|---|---|---|---|---|
| Limb | $O_4$ | 460–490 | 1a, 2a, 2c, 3, 4a | 2 | 1, 2, 3, 4, 5, 6, 7, 8 |
| Limb | $C_2H_2O_2$ | 420–439 and 447–465 | 1a, 2c, 3, 4a, 6 | 2 | 1 |
| | $C_2H_2O_2$ | 435–460 | 1b, 2b, 3, 4b, 4c, 6 | 3 | 4, 6, 8 |
| | $C_2H_2O_2$ | 430–460 | 1b, 2b, 3, 4b, 4c, 6 | 3 | 2, 3, 5, 7 |
| Nadir | $C_2H_2O_2$ | 435–460 | 1b, 2b, 3, 4b, 4c, 6 | 3 | 3, 4, 5, 6, 7, 8 |

Additional parameters for all spectral retrievals are (1) Offset spectrum; (2) Ring spectrum; (3) Ring spectrum multiplied by $\lambda^4$

(i): ACRDICON-CHUVA (1), OMO (2), EMeRGe-EU (3), WISE (4), EMeRGe-Asia (5), CoMet (6), CAFE-Africa (7), SouthTRAC (8).

mode, the instrument measures the atmospheric column density of the targeted gases below the aircraft with a rectangular foot print of $\sim 600\,\mathrm{m}$ cross-track and several $\mathrm{km}$ along-track (FOV $3.15° \times 0.38°$), depending on the flight altitude, cruising velocity (typically $\sim 200\,\mathrm{m\,s^{-1}}$ in the upper troposphere) and signal integration time (up to 1 min). Further, an IDS uEye camera (FOV $46°$) aligned with the Limb telescopes provides images of the sampled atmosphere at 1 Hz resolution.

More details of the instrument design, its major features, deployment on the HALO aircraft, the measurement method, the spectral retrieval and data processing can be found in Hüneke et al. (2017).

### 2.1.2 The spectral retrieval

The measured skylight spectra are analysed using the DOAS technique (Platt and Stutz, 2008). The retrieval details such as wavelength ranges, included trace gases and fitting parameters for each gas are provided in Tables 1 and 2.





The spectral retrieval settings of the Limb and Nadir glyoxal observations are based on the recent TROPOMI glyoxal
analysis from Lerot et al. (2021), in support of a better comparability of both data sets (see Sect. 4.2). Unlike the TROPOMI
glyoxal retrieval, the mini-DOAS measurements were not found to be significantly affected by changing $NO_2$ concentrations,
therefore only a single $NO_2$ absorption cross section (at a warmer temperature) is included in our analysis to account for the
tropospheric $NO_2$ absorption. A noteworthy difference to our recent glyoxal retrieval of the ACRIDICON-CHUVA data is the
use of a continuous fitting window ranging from 430 nm (or 435 nm for campaigns 4, 6, 8, and all Nadir observations) to
460 nm, instead of employing two simultaneous retrieval windows from 420 nm to 439 nm and 447 nm to 465 nm (Kluge
et al., 2020). The comparison of the spectral retrieval employing both wavelength ranges (i.e. when performing the spectral
analysis for each spectrum with both wavelength ranges) yields an improvement in the spectral residuum and signal to noise
ratio when using the continuous retrieval window, however with comparable dSCDs from both approaches. For the processing
of the data from different missions, minor adjustments to the retrieval settings (e.g. lower end of the analysed wavelength range
or temperatures of the included absorption cross sections) are applied when needed. Primarily, these adjustments are necessary
to compensate for mechanical modifications of the instrument and hence of the optical imaging (e.g. due to a fibre bundle
replacement, which causes changes in the lower wavelength limit of the spectrometers) as well as due to the largely changing
ambient conditions during the different research missions.

In particular for Nadir observations in a moist atmosphere and above the ocean, the DOAS analysis of glyoxal is challenging
due to its low optical density ($\sim 10^{-4}$) and proximity of the main absorption bands to the much stronger $7\nu$ absorption band of
water vapour (optical density on the order of $10^{-1}$ (Wei et al., 2019)). Therefore, for such observations, 5–150 consecutively
measured skylight spectra are co-added in order to minimise potential spectral interferences while optimising the signal to
noise, at the expense of enlarging the footprint of the affected measurements (as discussed in detail below).

Since the DOAS method infers a differential slant column density dSCD relative to a solar Fraunhofer reference spectrum
$SCD_{ref}$, the total slant column SCD has to be inferred from

$$SCD = SCD_{ref} + dSCD. \tag{1}$$

For the glyoxal retrieval, $SCD_{ref}$ is determined from a Limb measurement at high altitudes above which insignificant (or no)
glyoxal is expected, or if a sufficient flight altitude was not reached (EMeRGe-Europe and EmeRGe-Asia missions) by using
a reference from a different research flight which fulfills the above condition. For the retrieval of the $O_4$ extinction, $SCD_{ref}$ is
determined using a high resolution solar reference spectrum (Thuillier et al., 1998).

The mini-DOAS detection limit for glyoxal $dSCD_{dl}$ is $2.6 \cdot 10^{14}$ molec cm$^{-2}$ (Platt and Stutz, 2008). Depending on $SCD_{ref}$
and the related air mass factors (Sect. 2.1.3) and hence flight altitude, for Limb measurements, this results in a detection
limit ranging from 1 ppt (during clear skies, for a light path of about 100 km at maximum cruising altitude of the aircraft)
up to several 100 ppt (for very short line-of-sight photon paths, e.g. in dense clouds; Kluge et al. (2020)). For the Nadir
measurements, the typical detection limit $[VCD_{dl}]$ is $1.5 \cdot 10^{14}$ molec cm$^{-2}$, but depending on the flight altitude and cloud
cover and thus columnar air mass factor, it can be as low as $6.7 \cdot 10^{13}$ molec cm$^{-2}$ or as large as $2.6 \cdot 10^{14}$ molec cm$^{-2}$.





### 2.1.3  Retrieval of concentrations (Limb) and vertical column densities (Nadir)

The following section discusses the conversion of the inferred SCDs to mixing ratios (Limb measurements) and to total vertical column densities (Nadir measurements).

The Limb measured slant column densities $SCD_{limb}$ are converted into trace gas mixing ratios using the $O_4$-scaling method, as described in detail in Hüneke (2016); Hüneke et al. (2017); Stutz et al. (2017); Werner et al. (2017); Kluge et al. (2020). Accordingly, the concentration of a trace gas [X] at flight altitude j is inferred from $SCD_{limb,X}$ by comparing the measured optical depth $SCD_{limb,O_4}$ with the calculated clear sky extinction $[O_4]_j$ and taking into account radiative transfer-based correction factors $\alpha_{X_j}$ and $\alpha_{O_{4,j}}$) to quantify the optical characteristics (e.g. aerosol and cloud cover) of the radiative transfer during

each single measurement:

$$[\mathrm{X}]_j = \frac{\alpha_{X_j}}{\alpha_{O_{4,j}}} \cdot \frac{SCD_{limb,X}}{SCD_{limb,O_4}} \cdot [O_4]_j. \tag{2}$$

This approach is justified based on the equivalence theorem in optics (Irvine, 1963), which states that for a given wavelength, the photon path length distribution and hence the mean photon path lengths are the same for weak absorbers with similar atmospheric distributions. Evidently, this criterion is reasonably well approximated when using $O_4$ as scaling gas for all trace

gases with sources at the ground and sinks in the troposphere, such as $C_2H_2O_2$. The remaining differences in the profile shapes of the target (X) and scaling gas ($O_4$) and their centre wavelengths of absorption are accounted for by the so called $\alpha$-factors. The $\alpha$-factors express the fraction of absorption within the line-of-sight at the measurement altitude relative to the total atmospheric absorption, which may differ slightly for the gas of interest compared to the scaling gas due to their different profile shapes and absorption wavelengths. The $\alpha$-factors are simulated using the McArtim radiative transfer model (Deutschmann

et al., 2011). For the simulation, in a first step a priori profiles of glyoxal are used based on previous mini-DOAS measurements in similar atmospheric conditions (clean and polluted terrestrial air, remote marine air et cetera), and subsequently iterated until convergence is achieved (see below). In the following, the robustness of the iterative approach is described and tested. In a first step, the RT model is run using an a priori glyoxal profile obtained during previous missions and assuming an exponentially decaying profile above the maximum flight altitude. Secondly, a modelled a priori glyoxal profile is used based on simulations

from the global chemical transport model MAGRITTE (Model of Atmospheric composition at Global and Regional scales using Inversion Techniques for Trace gas Emissions). This model is used e.g. to calculate air mass factors for the TROPOMI retrieval (e.g. Bauwens et al. (2016); Lerot et al. (2021)). Here, the simulated MAGRITTE glyoxal a prioris are averaged along a research flight track over central Europe on 15 May 2018. Both a prioris -simulated and measured- show notable differences in their absolute mixing ratios as well as in their profile shape (Fig. 1, panels a and b). In the following, iterations of the

radiative transfer simulation are performed for the flight on 15 May 2018 using consecutively resulting profiles from both approaches as new a priori profiles (Fig. 1, panels a and b). Evidently, even strongly diverging a priori profiles converge well after the first iteration to a common profile constrained by the measured SCDs within the error margins. After four iterations, no notable changes are discernible in the obtained a priori profiles as well as in the resulting mixing ratios (Fig. 1, panel c), thus demonstrating the robustness of the iteration with respect to the a priori profile assumptions. This insensitivity of the resulting

profiles towards the assumed a priori profile is a direct consequence of the applied scaling method, because all a priori profiles



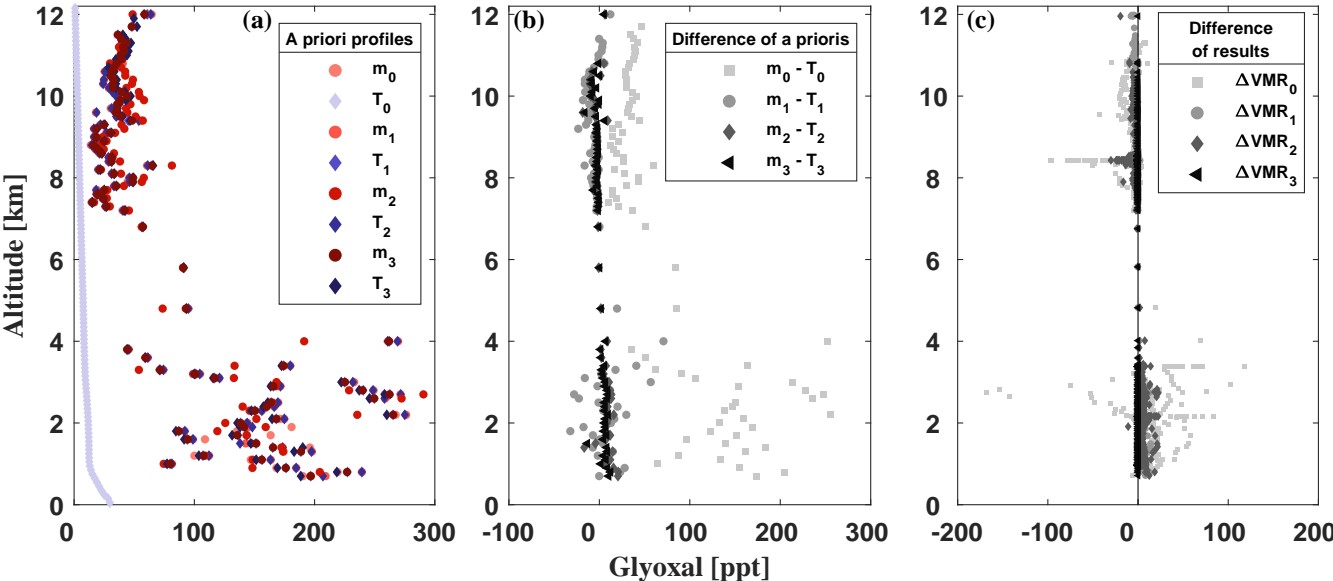

**Figure 1.** Iterations of different a priori glyoxal profiles in $\alpha$-factor calculations, shown here for the HALO flight on 15 May 2018 (CoMet mission). Starting a priori profiles are a mean glyoxal profile inferred from previous mini-DOAS measurements in polluted continental air ($m_0$), and an averaged glyoxal profile simulated by the MAGRITTE model ($T_0$) for a TROPOMI overpass over Europe at 13:30 UTC on 15 May 2018 (Lerot et al., 2021). These a prioris are used for the first iteration in the $\alpha$-factor calculations. The resulting profiles are then used for the following i iterations to obtain the profiles $m_i$ and $T_i$, respectively (panel a). The differences $m_i - T_i$ of the a priori profiles after each iteration step are shown in panel (b). The differences in the resulting mixing ratios from each iteration step (based on the respective $\alpha$-factor simulation) are shown in panel (c).

of the target gas are evaluated relative to the scaling gas (in this study $O_4$), of which the atmospheric absorption is measured. Without iteration, the scaling method is therefore moderately sensitive to the relative differences in the assumed profile shapes (Fig. 1, panel b), but not to the absolute concentrations assumed in the a priori profiles. As demonstrated above, this remaining sensitivity can sufficiently be reduced by performing a second iteration of the radiative transfer model when calculating the

$\alpha$-factors.

For the Nadir measurements, the measured slant column densities $SCD_{nadir}$ of the trace gas X are converted into vertical column densities $VCD_X$ according to

$$VCD_{X,j} = SCD_{nadir,X,j} \cdot \frac{\sum [X]_j \cdot z_j}{\sum [X]_j \cdot z_j \cdot B_j}, \quad (3)$$

using box air mass factors $B_j$ simulated by the McArtim model for each layer j of thickness $z_j$ and equal a priori concentrations

$[X]_j$ in each altitude bin j as used for the Limb measurements. For the measurements at lower flight altitudes, $VCD_{X,j}$ does not correspond the total vertical column of glyoxal, as only the fraction of the absorption below the aircraft is measured. Therefore,





for the comparison of mini-DOAS Nadir measurements with respect to TROPOMI satellite observations, only VCDs measured above 8 km flight altitude are used, for which over 95 % of the atmospheric glyoxal is expected to be located below the aircraft.

The footprint (Nadir) or average atmospheric volume (Limb) analysed from each spectrum can be approximated based on
the aircraft displacement (vertical and horizontal) during the spectrum integration time as well as the telescope's rectangular field of view ($0.5°$ vertical, $3.15°$ horizontal) and the mean light path length. The latter two processes are considered in the evaluation and respective RT calculations.

In the Nadir viewing geometry, for flight altitudes above 8 km in a moderately humid to dry atmosphere, the typical footprint for a signal integration time of 14 s is $4.2 \times 0.6$ km$^2$. For flights in a moist atmosphere or over the pristine oceans, occasionally
up to 150 spectra were co-added (South Atlantic measurements), thus extending the median Nadir along-track resolution in extreme cases up to 230 km. The median Nadir along-track resolution over the South Atlantic is 32 km. While this spectral co-adding may enlarge the footprint, it favors the detection limit which is helpful for monitoring glyoxal at low VCDs, i.e. far away from distinct sources such as over the open ocean. In the other regions, where less or no spectral co-adding was necessary, the median Nadir along-track resolution is 4 km (Europe, Mediterranean and East China Sea, Tropical and North Atlantic).

## 2.2  TROPOspheric Monitoring Instrument (TROPOMI)

The Nadir-viewing Tropospheric Monitoring Instrument (TROPOMI) was launched on 13 October 2017 on board the Copernicus Sentinel-5 Precursor satellite platform on an early afternoon sun-synchronous orbit with a local equator crossing-time of 13:30 LT (Veefkind et al., 2012). It measures solar irradiance and Earthshine radiance spectra in the ultraviolet, visible, and near and short-wave infrared spectral ranges with a spectral resolution of about 0.5 nm (0.25 nm in the short-wave in-
frared). TROPOMI is a push-broom instrument with a large swath of about 2600 km, which provides a daily global coverage of the Earth's atmosphere with an unprecedented spatial resolution (up to $3.5 \times 5.5$ km$^2$). Those measurements allow retrieving vertical column densities (i.e. vertically integrated concentrations) for a series of key trace gases but also deriving crucial information on clouds and aerosols. Since the beginning of the mission, TROPOMI has therefore provided essential information for air quality and climate applications and in particular via the distribution of operational products for ozone ($O_3$), nitrogen diox-
ide ($NO_2$), carbon monoxide (CO), sulfur dioxide ($SO_2$), methane ($CH_4$) and formaldehyde ($CH_2O$). In addition, TROPOMI measurements can be exploited to infer information on other species, including glyoxal as recently demonstrated by Lerot et al. (2021). More details on the satellite glyoxal retrievals are given in Sect. 3.2.

## 2.3  ECHAM/MESSy Atmospheric Chemistry (EMAC) model

The ECHAM/MESSy Atmospheric Chemistry (EMAC) model is a numerical chemistry and climate simulation system that
includes submodels describing tropospheric and middle atmospheric processes and their interaction with oceans, land, and human influences (Jöckel et al., 2010). It uses the second version of the Modular Earth Submodel System (MESSy2) to link multi-institutional computer codes. The core atmospheric model is the fifth-generation European Centre Hamburg general circulation model (ECHAM5; Roeckner et al., 2003).





## 3 Measurements and simulations

### 3.1 The mini-DOAS measurements

This study presents airborne glyoxal measurements from eight different research missions performed in the period between fall 2014 and fall 2019. In total, 72 research flights were performed with the German research aircraft HALO operated by the Deutsches Zentrum für Luft- und Raumfahrt (DLR) in Oberpfaffenhofen, Germany. The scientific flights covered a wide range of geographic areas reaching from the southern tip of South America and Western Antarctica, over the Amazon rainforest,

the Tropical and North Atlantic, and Europe to the East China Sea, Taiwan, and the southern Japanese islands (Fig. 2). The study combines Nadir and Limb measurements of glyoxal performed during the following scientific missions: (1) Aerosol, Cloud, Precipitation, and Radiation Interactions and Dynamics of Convective Cloud Systems (ACRDICON-CHUVA) over the Amazon in fall 2014 (Wendisch et al., 2016), (2) Oxidation Mechanism Observations (OMO) over the Mediterranean and Arabian Sea and Indian Ocean in summer 2015 (Lelieveld et al., 2018), (3) the Effect of Megacities on the Transport and

Transformation of Pollutants on the Regional to Global Scales in Europe (EMeRGe-EU) in summer 2017 (Andrés Hernández et al., 2022), (4) Wave-driven ISentropic Exchange (WISE) over the North Atlantic and Europe in fall 2017, (5) the Effect of Megacities on the Transport and Transformation of Pollutants on the Regional to Global Scales in Asia (EMeRGe-Asia) in early spring 2018, (6) the Carbon Dioxide and Methane Mission (CoMet) over Europe in late spring 2018 (Fix et al., 2018), (7) Chemistry of the atmosphere: African Field Experiment (CAFE-Africa) over the Tropical Atlantic and West Africa in summer

2018, and (8) Transport and Composition of the Southern Hemisphere UTLS (SouthTRAC) in Patagonia/Western Antarctica in fall 2019 (Rapp et al., 2021). Detailed descriptions of each mission, the deployed instruments and research objectives can be found in the respective mission publications.

ACRDICON-CHUVA (main operation base Manaus, Brazil), OMO (operation base Paphos, Cyprus), EMeRGe-EU and CoMet (both operated from the home base in Oberpfaffenhofen, Germany) were predominantly conducted over land and the

adjacent coastal regions. Most research flights from the other missions focused on remote marine measurements over the South Pacific and South Atlantic/Wedell Sea (SouthTRAC, operation base Rio Grande, Argentina), the Tropical Atlantic (CAFE-Africa, operation base Sal, Cape Verde), the North Atlantic (WISE, operation base Shannon, Ireland), and the East China Sea (EMeRGe-Asia, operation base Tainan, Taiwan; Fig. 2).

During the eight research missions, air masses of different origins and compositions, and thus largely different glyoxal

sources and concentrations were probed. This included (1) pristine marine air, (2) pristine continental air, (3) biomass burning affected air of different ages, and (4) air affected by fresh or aged anthropogenic emissions.

(1) Pristine marine air was primarily probed over the North, Tropical, and South Atlantic during the missions WISE in 2017, CAFE-Africa in 2018, and SouthTRAC in 2019. During all three missions, long flight sections took place in the upper troposphere over the remote oceans, with occasional dives into the marine boundary layer as well as to the airports of Shannon

(Ireland), Sal (Cape Verde), or Rio Grande (Argentina). Combined, the three missions covered the Atlantic from Iceland and Scandinavia in the north, over the Azores and the equatorial latitudes, down to the South Atlantic, South-East Pacific and the Wedell Sea around the Antarctic Peninsula.





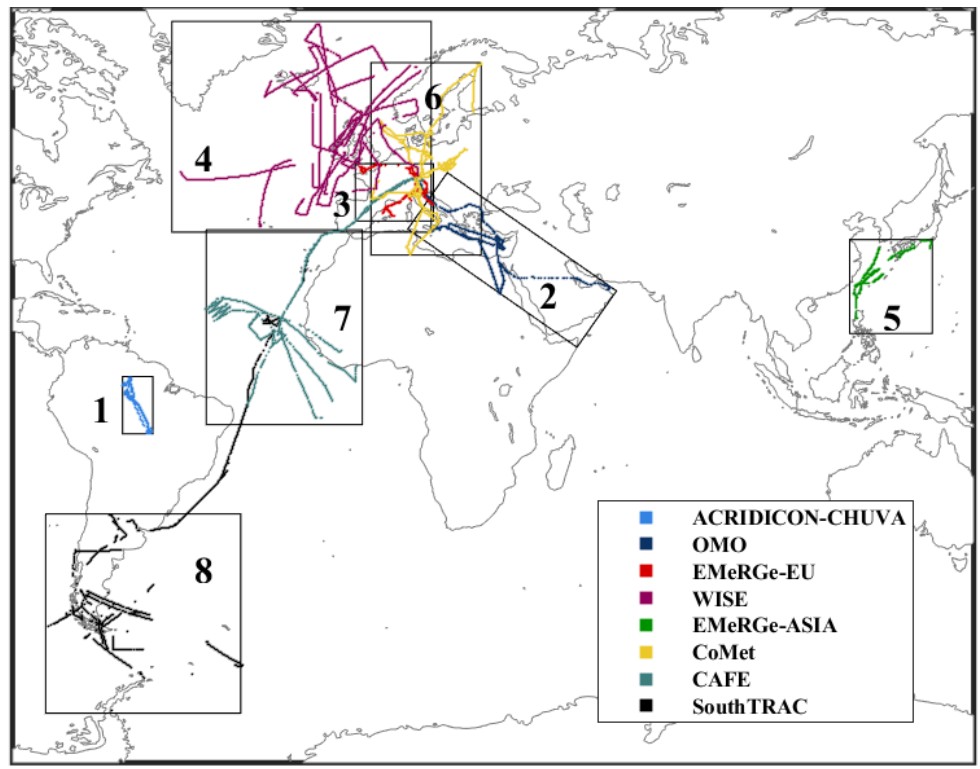

**Figure 2.** The different geographical regions and the flight tracks covered during the eight missions. For simplicity, the missions are numberd according to their chronological order with ACRDICON-CHUVA (1), OMO (2), EMeRGe-EU (3), WISE (4) EMeRGe-Asia (5), CoMet (6), CAFE-Africa (7), SouthTRAC (8). The operational bases for the individual missions were Manaus (Brazil) for ACRDICON-CHUVA, Paphos (Cyprus) for OMO, Oberpfaffenhofen (Germany) for EMeRGe-Europe and CoMet, Shannon (Ireland) for WISE, Tainan (Taiwan) for EMeRGe-Asia and Rio Grande (Argentina) for SouthTRAC. Only flight sections with mini-DOAS measurements are shown.

(2) The most pristine terrestrial air masses were found over Patagonia, south Argentina and northern Antarctica during the SouthTRAC mission in austral spring 2019, with vertical profiling mostly performed during landward ascents from and
descents into the Rio Grande and Buenos Aires airports (Argentina).

(3) Air masses affected by biogenic emissions and biomass burning were intensively probed over the Amazon during the ACRDICON-CHUVA mission in fall 2014 and the measurements of $CH_2O$, $C_2H_2O_2$ and $C_3H_4O_2^*$ have been discussed in detail in Kluge et al. (2020). Additionally, aged biomass burning plumes advected from continental Africa and Brazil were detected during several flights of the CAFE-Africa mission over the Tropical Atlantic as well as during several transfer flights of
the SouthTRAC mission from Europe along the Brazilian coastline towards Patagonia (Argentina). On 12 Nov. 2019, biomass burning affected air masses from the Australian bushfires in austral spring and summer 2019 were detected over the Drake Passage around 57° S and 67° W in the upper troposphere (e.g. Kloss et al. (2021)).





(4) Air masses affected by recent anthropogenic emissions were probed over the European continent and its northern islands (mostly Ireland and Great Britain) as well as over the adjacent marine regions (North, Baltic and Mediterranean Sea) during the missions EMeRGe-EU in summer 2017, WISE in fall 2017, CoMet in spring 2018, and over Taiwan and the East China Sea during EMeRGe-Asia in spring 2018. The research flights targeted the emissions of major European and Asian cities, e.g. London, Paris, Rome, Marseille, Barcelona, Manila, and Osaka as well as emissions from industrial areas (e.g. the Ruhr Valley and the Upper Silesian Coal Basin). During the EMeRGe missions, the research flights focused on fresh pollution plumes in the planetary and marine boundary layer, whereas the CoMet flights provided measurements in the free and upper troposphere over the different cities, Scandinavia, the Baltic and the Mediterranean Sea. In addition to these detailed continental measurements, observations of the northern European coastal regions were completed during the WISE mission, including Ireland, Great Britain, and Iceland with a special focus on fresh anthropogenic emissions over the Irish Sea and the North Channel.

Air masses affected by aged pollution were probed in all regions, especially along the coast of the Mediterranean Sea, Egypt, the Arabian Peninsula and Arabian Sea during the OMO mission in summer 2015 (Lelieveld et al., 2018), as well as in the outflow of mainland China, Japan, Korea, and the Philippines over the East China Sea during EMeRGe-Asia, and during several flights along the Brazilian coastline. During OMO, measurements were predominantly conducted in the upper troposphere, but also included profiling into the boundary layer, i.e. over Cyprus and Bahrain. Due to operational reasons, during EMeRGe-Asia profiling from the free troposphere into the boundary layer could only be performed over Taiwan, south of Osaka (Japan), and Manila (Philippines), whereas free and upper tropospheric air was probed during the flight sections between Taiwan and south Japan.

Glyoxal measurements are sometimes only available for a portion of the total flight time. This is mostly a result of (1) flight sections of recorded spectra with unfavorable sunlight conditions causing an over-saturation of the CCD detector (e.g. when flying in and adjacent to bright clouds), (2) research flights during the night (e.g. during SouthTRAC), or (3) spectrometer temperatures being above 3° C preventing a stable spectral imaging (e.g. during EMeRGe-Asia).

## 3.2 TROPOMI measurements

Glyoxal tropospheric vertical columns can be retrieved from the satellite TROPOMI observations by exploiting its absorption bands in the visible spectral range using a DOAS approach. Here we use the TROPOMI glyoxal product recently developed by BIRA-IASB (Lerot et al. (2021)). The algorithm consists of three consecutive steps.

(1) Glyoxal slant column densities are derived from a DOAS spectral fit of the measured optical depths performed in the window 435–460 nm. In addition to glyoxal, absorption cross-sections are included to account for spectral signatures from ozone, $NO_2$, $O_2-O_2$, water vapour, liquid water, inelastic scattering, and scene brightness heterogeneity. The glyoxal optical depth is generally weak, which makes the product noisy and sensitive to spectral interferences. The level of noise can be reduced by averaging observations in space and/or time. An empirical correction is also applied to the glyoxal slant columns in case of extreme $NO_2$ absorption to reduce the impact of the resulting misfit.

(2) The slant column densities are converted into vertical column densities via air mass factor, which are computed by combining weighting functions and a priori glyoxal profile shapes (Palmer et al. (2001)). The a priori profiles are provided



over land by the CTM MAGRITTE, an update of the IMAGES model (Müller and Brasseur (1995); Bauwens et al. (2016)), while a single profile measured in 2012 over the Pacific Ocean (Volkamer et al. (2015)) is used over oceanic regions. The impact of clouds is neglected and a stringent filter is applied to reject cloud-contaminated scenes.

(3) A background-correction based on measurements in the remote Pacific Ocean is applied and aims at reducing the presence of biases originating from interferences with spectral signatures from other absorbers or calibration limitations. For example, this procedure is designed to reduce an identified row-dependent bias. Owing to the empirical nature of the correction, some remnants of this bias may nevertheless still be visible when maintaining the daily time resolution. The quality of this TROPOMI glyoxal product has been assessed in Lerot et al. (2021) based on comparisons with other glyoxal satellite data sets

produced with a similar algorithm and with a series of MAX-DOAS data sets. A high level of consistency is found between the satellite data sets (within 20 %) and high correlation coefficients are found between TROPOMI and MAX-DOAS data sets, indicating that the glyoxal variability observed from space and ground agree well. Enhanced glyoxal columns are also observed by TROPOMI over equatorial oceans. The respective contributions of physical processes and spectral interferences to those elevated columns have remained unclear for years. Additional comparisons with independent measurements such as

those presented in this study are therefore crucial to provide insight into this issue.

### 3.3 EMAC simulations

In the present study, EMAC (ECHAM5 version 5.3.02, MESSy version 2.55.0) is used at T63L90MA resolution, i.e. with a spherical truncation of T63 (corresponding to a quadratic Gaussian grid of approximately 1.875° by 1.875° in latitude and longitude) with 90 vertical hybrid pressure levels up to 0.01 hPa. In order to reproduce the actual day-to-day meteorology in

the troposphere, the dynamics have been weakly nudged (Jöckel et al., 2006) towards European Centre for Medium-Range Weather Forecasts (ECMWF) Reanalysis v5 (ERA5; Hersbach et al., 2020) data.

Atmospheric gas-phase chemistry is calculated employing the Module Efficiently Calculating the Chemistry of the Atmosphere (MECCA; Sander et al., 2019) using the gas-phase Mainz Organic Mechanism (MOM) recently evaluated by Pozzer et al (2022). MOM contains an extensive oxidation scheme for isoprene (Taraborrelli et al., 2009, 2012; Nölscher et al., 2014;

Novelli et al., 2020), monoterpenes (Hens et al., 2014), and aromatics (Cabrera-Perez et al., 2016). In addition, comprehensive reaction schemes are considered for the modelling of the chemistry of $NO_x$, $HO_x$, $CH_4$, and anthropogenic linear hydrocarbons. VOCs are oxidised by OH, $O_3$, and $NO_3$, whereas $RO_2$ reacts with $HO_2$, $NO_x$, and $NO_3$ and undergoes self- and cross-reactions. All in all, MOM considers 43 primarily emitted VOCs and represents more than 600 species and 1600 reactions (Sander et al., 2019).

The SCAVenging submodel (SCAV; Tost et al., 2006) is used to simulate the removal of trace gases and aerosol particles by clouds and precipitation. SCAV calculates the transfer of species into and out of rain and cloud droplets using Henry's law in equilibrium, acid dissociation equilibria, oxidation–reduction reactions, heterogeneous reactions on droplet surfaces, and aqueous-phase photolysis reactions. In its basic setup, SCAV is used to calculate EMAC's standard aqueous-phase mechanism including the representation of more than 150 reactions and even includes a simplified degradation scheme of methane oxida-



tion products (Tost et al., 2007). The aqueous-phase representation of formaldehyde ($CH_2O$) is updated following Franco et al. (2021).

Anthropogenic emissions are based on the Emissions Database for Global Atmospheric Research (EDGAR, v4.3.2; Crippa et al., 2018), and vertically distributed following Pozzer et al. (2009). The Model of Emissions of Gases and Aerosols from Nature (MEGAN; Guenther et al., 2006) is used to model biogenic VOC emissions. Global isoprene emissions are scaled to the best estimate of Sindelarova et al. (2014). Biomass burning emission fluxes are calculated using the MESSy submodel BIOBURN, which calculates these fluxes based on biomass burning emission factors and dry matter combustion rates. For the latter, Global Fire Assimilation System (GFAS) data are used, which are based on satellite observations of fire radiative power from the Moderate Resolution Imaging Spectroradiometer (MODIS) satellite instruments (Kaiser et al., 2012). The biomass burning emission factors for VOCs are based on Akagi et al. (2011), excluding direct glyoxal emissions. The MESSy submodel AIRSEA is used to represent the air-sea exchange of isoprene and methanol following Pozzer et al. (2006).

The applied EMAC modelling setup is representable for recent studies that focused on a detailed representation of VOCs (e.g. by using MOM to represent gas-phase chemistry; Novelli et al. (2020); Rosanka et al. (2020); Taraborrelli et al. (2021); Franco et al. (2021); Rosanka et al. (2021a); Pozzer et al. (2022). Using MOM in EMAC at a resolution of T63L90MA comes at a high computational demand. Therefore, we perform an EMAC simulation focusing on the years 2017, 2018, and 2019, which covers most airborne missions except ACRIDICON-CHUVA in 2014 and OMO in 2015. For these missions, simulation results of the year 2017 are used for the climatological comparison with observational data. All simulations were performed at the Jülich Supercomputing Centre using the Jülich Wizard for European Leadership Science (JUWELS) cluster (Krause, 2019).

## 4 Observations

Based on the data collected during all research missions, atmospheric glyoxal profiles are inferred in (1) recently polluted air both over continents (Europe, South America, and South-East Asia) as well as adjacent marine regions (e.g. Mediterranean Sea, Irish Sea, and East/South China Sea), (2) biomass burning affected air (mostly off-coast Africa and South America), and (3) pristine marine air (South, Tropical, North Atlantic). The following section firstly discusses the inferred vertical profiles of glyoxal and compares them to previous measurements, before the Nadir VCDs and integrated profiles are inter-compared with collocated TROPOMI glyoxal VCD measurements. Finally, the airborne measurements of glyoxal concentrations and VCDs are compared to respective simulations from the EMAC model.

### 4.1 Glyoxal profiles

Figure 3 provides an overview of all glyoxal profiles inferred for the eight investigated regions. This includes glyoxal profiles measured over Patagonia and the Amazon rainforest (Fig. 3, panel a), East China Sea and Taiwan (panel b), the European continent and its northern islands (panel c), the South Atlantic (panel d), the Tropical Atlantic (panel e), the North Atlantic (panel f), and the Mediterranean Sea (panel g). The corresponding flight tracks as a function of altitude are shown in Fig. 4





**Table 3.** Observed median, standard deviation and maximum (in brackets) glyoxal mixing ratios [ppt] as a function of different geographic regions, air mass types, and altitude ranges. If available, chemical markers are used for the attribution of the air masses to the different regimes: Polluted air masses are differentiated into fresh or aged biomass burning ($P_{BB-fresh/aged}$ or $P_{BB}$, in cases where the plume age is not known), anthropogenic pollution ($P_{anthr.}$), and mixed pollution ($P_{mix}$). Further categories consist of air masses rich in bVOC emissions (labelled biogenic, e.g. the tropical rainforest) as well as pristine air.

| Region | Air mass type | Glyoxal [ppt] | | |
| --- | --- | --- | --- | --- |
| | | Lower troposphere 0–3 km | Free troposphere 3–8 km | Upper troposphere 8–15 km |
| **South Atlantic** | pristine | $10 \pm 25$ (139) | $3 \pm 28$ (84) | $3 \pm 5$ (36) |
| | $P_{BB-aged}$ | | $33 \pm 9$ (45) | $31 \pm 29$ (187) |
| **Tropical Atlantic** | pristine+$P_{BB-aged}$ | $44 \pm 64$ (382) | $13 \pm 27$ (185) | $1 \pm 9$ (246) |
| | $P_{BB-aged}$ | $58 \pm 16$ (94) | $41 \pm 37$ (168) | $5 \pm 101$ (3192) |
| **North Atlantic** | pristine | $19 \pm 30$ (144) | $13 \pm 13$ (51) | $5 \pm 11$ (191) |
| | $P_{anthr.}$ | | | $53 \pm 301$ (2970) |
| **Mediterranean Sea** | $P_{mix}$ | $279 \pm 144$ (845) | $20 \pm 64$ (309) | $5 \pm 9$ (87) |
| | $P_{BB-aged}$ | $126 \pm 15$ (147) | | |
| | $P_{anthr.}$ | $182 \pm 153$ (570) | | |
| **South America** | pristine | $23 \pm 39$ (223) | $24 \pm 33$ (204) | $5 \pm 9$ (119) |
| | $P_{BB-aged}$ | | $96 \pm 5$ (102) | $3 \pm 66$ (311) |
| | rainforest | $87 \pm 45$ (476) | $36 \pm 78$ (593) | $3 \pm 12$ (33) |
| **East China Sea** | $P_{mix}$ | $75 \pm 68$ (323) | $11 \pm 27$ (340) | $2 \pm 43$ (418) |
| | $P_{BB-aged}$ | $52 \pm 50$ (301) | $29 \pm 28$ (109) | $6 \pm 53$ (234) |
| | $P_{anthr.}$ | $77 \pm 67$ (299) | $15 \pm 62$ (474) | |
| **Continental Europe** | $P_{mix}$ | $51 \pm 118$ (580) | $14 \pm 34$ (441) | $6 \pm 17$ (493) |
| | $P_{BB}$ | | $12 \pm 1$ (12) | |
| | $P_{anthr.}$ | $399 \pm 121$ (547) | $204 \pm 104$ (277) | |

(panels a, d, g, j, and m) together with the along track inferred glyoxal mixing ratios (panels b, e, h, k, and n) and glyoxal VCDs (panels c, f, i, l, and o).

As can be seen in Fig. 4, lower altitudes were mostly probed in the vicinity of potential emission sources (e.g. large popula-
tion centres and biomass burning events), whereas the remote oceans were predominantly probed from the upper troposphere (Fig. 4 and Sect. 4.2). This sampling was motivated by (1) the mission objectives, which often differed from those necessary for in depth glyoxal monitoring, (2) flight track restrictions, which prohibited some of the intended soundings (e.g. over the remote



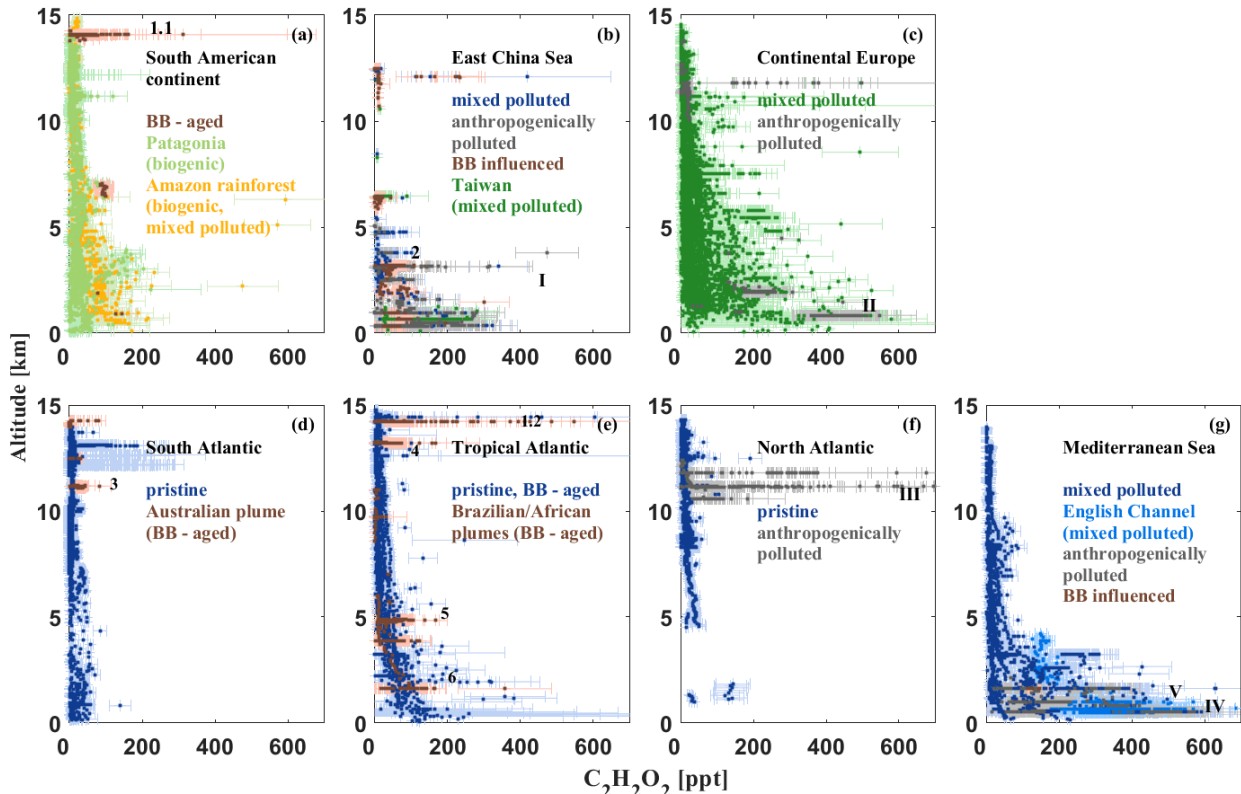

**Figure 3.** Vertical profiles of glyoxal in the different regions: South American continent in fall 2014 and austral spring 2019 (panel a); Taiwan and the East China Sea in spring 2018 (panel b); the European continent in summer and fall 2017 and 2018 (panel c); the South Atlantic in austral spring 2019 (panel d); the Tropical Atlantic and around the Cape Verde Islands in summer 2018 and austral spring 2019 (panel e); the North Atlantic and Irish Sea in fall 2017 (panel f); and the Mediterranean Sea, English channel and North Sea in summer 2015, 2017, and 2018 (panel g). The various marine environments are indicated in blue, terrestrial in green, and rainforest in yellow, while perturbations due to biomass burning (BB) plumes are indicated in brown and city plumes in grey. The Arabic numbers denote encounters with biomass burning plumes and the Roman numerals with plumes of anthropogenic emissions (mostly city plumes). The three largest glyoxal plumes observed are plume 1.2 (panel e) with up to 3192 ppt, plume III with 2970 ppt (panel f), and plumes IV and V with 845 ppt and 774 ppt, respectively (panel g). All four plumes are not shown in full scale for better comparability with the other profiles. For better visibility of the different profile shapes, occasional negative measurements are not shown (compare to Fig. 10).

South Atlantic), (3) the required aircraft tracks with narrow curves near airports which prevented a reliable Limb sounding, and (4) instrument malfunctions. Consequently, the marine observations in the lower troposphere over the South and Tropical Atlantic as well as over the Mediterranean Sea, contain a larger fraction of coastal rather than remote ocean soundings. Also the vertical soundings over the Mediterranean Sea and North Atlantic were mostly performed near larger pollution sources (e.g. Marseilles, Barcelona, and Shannon), rather than over the remote sea, leading to a larger fraction of pollution-affected observations in the planetary boundary layer compared to higher altitudes. However, this bias is expected to be small for ver-



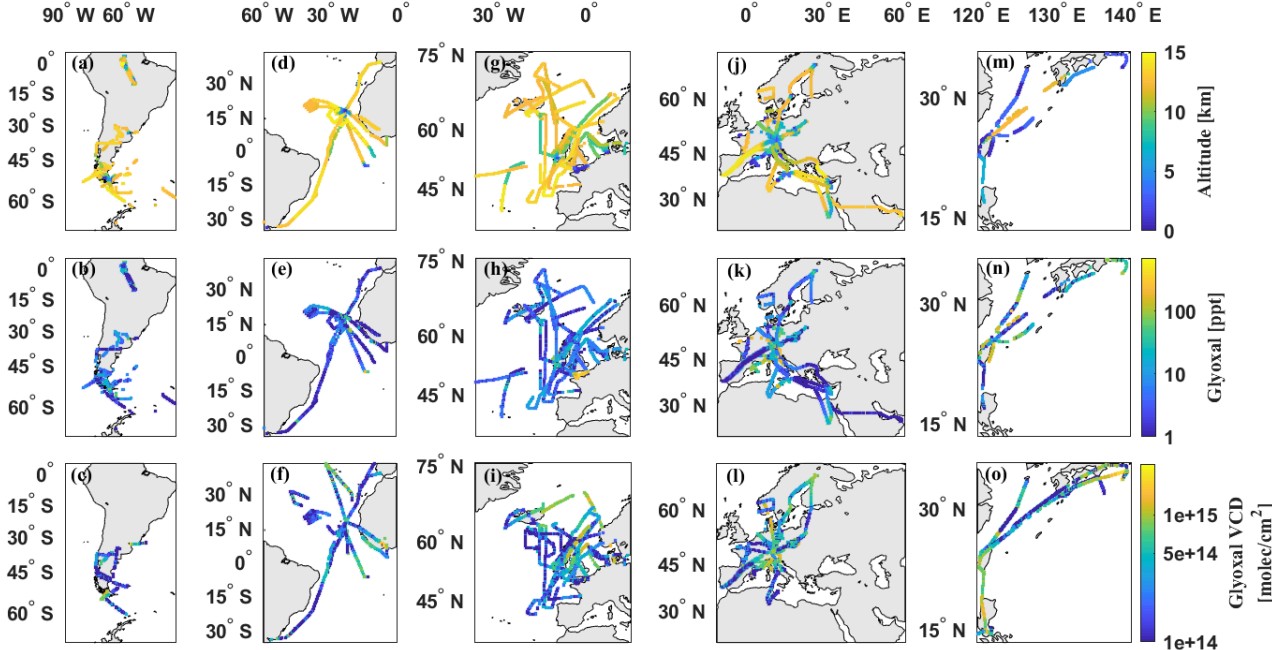

**Figure 4.** Flight trajectories, measured glyoxal mixing ratios, and VCDs are displayed in the top, middle and bottom rows, respectively, from over South America (panels a, f, and k), the Tropical Atlantic (panels b, g, and l), the North Atlantic (panels c, h, and m), Continental Europe (panels d, i, and n), and South-East Asia (panels e, j, and o). Note the logarithmic colour scale in the two lower rows of the panels.

tical soundings in regions of widely polluted air (e.g. over polluted continental areas) and the East China Sea, where vertical
profiles were also inferred over the remote ocean.

Nevertheless, of these observation related limitations, the glyoxal profiles provide valuable information on the background of glyoxal in various environments around the globe including marine (Fig. 3, blue colour) and terrestrial regions (Fig. 3, green colour) and the Amazon rainforest (Fig. 3, yellow colour) as well as perturbations due to biomass burning (BB) plumes (Fig. 3, brown colour) and city plumes (Fig. 3, grey colour).

The attribution of the different air masses to the respective emission types is based on markers of typical chemicals emitted by anthropogenic activities and biomass burning. For the EMeRGe-EU and EMeRGe-Asia missions, benzene ($C_6H_6$), isoprene ($C_5H_8$), and acetonitrile ($CH_3CN$) are used to detect anthropogenic pollution or biomass burning affected air masses (Andrés Hernández et al., 2022; Krüger et al., 2022). Biomass burning plumes are further identified based on simultaneous measurements of CO (Tadic et al., 2017) and black carbon (Holanda et al., 2020) during CAFE-Africa, and of CO (Müller
et al., 2015; Kunkel et al., 2019), peroxyacetyl nitrate (PAN), ethane ($C_2H_6$), formic acid (HCOOH), methanol ($CH_3OH$), ethylene ($C_2H_4$) (Johansson et al., 2022), and acetylene ($C_2H_2$) (S. Johansson, private communication, Feb. 2022) during the SouthTRAC mission, and by visual inspection as well as air mass back trajectory calculations for the ACRIDICON-CHUVA mission (Kluge et al., 2020).





Evidently, not all of the glyoxal observations can unambiguously be assigned to one of the four encountered air mass types (pristine, biogenic, polluted, or biomass burning affected) as described above, since either no markers for the different regimes were measured during individual flights or entire missions (e.g. CoMet) or a mixture of polluted air masses from different sources was probed, e.g. over continental Europe, the Mediterranean Sea, Taiwan and the East China Sea (Lelieveld et al., 2018; Andrés Hernández et al., 2022).

Table 3 provides an overview of glyoxal mixing ratios (median, standard deviation and maximum) inferred in the distinctive regimes in the lower, middle and upper troposphere in the different global regions. As expected, most glyoxal is observed in the lower and middle troposphere over Europe, the Mediterranean Sea and Eastern Asia, and the smallest concentrations are found over pristine marine (South Atlantic) and terrestrial regions (Patagonia). In the upper troposphere, glyoxal mixing ratios are generally very small (a few ppt) in all regions, if not affected by biomass burning or other emission plumes.

Overall, the comparison of our glyoxal measurements with previous findings from varying altitudes, regions and seasons around the globe shows a reasonable good agreement.

**(1) Pristine marine air masses** were mostly probed over the South Atlantic, with median mixing ratios in the lower, middle, and free troposphere of 10 ppt, 3 ppt, and 3 ppt, respectively (Fig. 3, panel d and Fig. 4, panels a–c). Over the South Atlantic, glyoxal mixing ratios larger than 100 ppt are exclusively observed in aged biomass burning plumes or near the South American coastline below 3 km altitude, where the enhancements very likely result from respective continental outflow of glyoxal and its precursors. Comparable glyoxal mixing ratios of 19 ppt (median) are observed in the planetary boundary layer over the North Atlantic, with slight glyoxal enhancements predominantly observed during ascents and descents into the Shannon airport (Ireland; Fig. 4, panels g–i). Also in the middle and free troposphere, the range of observed glyoxal mixing ratios above the pristine South and North Atlantic compares well if not affected by specific emission events (biomass burning for the South and mostly anthropogenic emissions for the North Atlantic, see Fig. 3, panels d and f). Glyoxal observations in the remote marine free troposphere and in particular in the free troposphere over the North Atlantic are still sparse, and thus deserve further investigations.

In the pristine marine boundary layer, our glyoxal measurements compare well with observations by Mahajan et al. (2014), who found average glyoxal mixing ratios of typically 25 ppt (upper limit 40 ppt) based on 10 oceanic cruises over the open oceans in different parts of the world. For the South Pacific boundary layer, Lawson et al. (2015) reported glyoxal mixing ratios ranging from 7 to 23 ppt, which is slightly smaller than our median observation in the boundary layer over the Tropical Atlantic of 44 ppt.

In fact, compared to the measurements in pristine marine air (e.g. over the South Atlantic), glyoxal over the Tropical Atlantic below 3 km is found to be on average 4 times larger (Fig. 3, panel e and Fig. 4, panels d–f). These observations of moderately elevated glyoxal in the marine lower troposphere in the tropics support previous findings of elevated glyoxal elsewhere in the remote marine tropics, e.g. in the lower atmosphere over the East Pacific (Sinreich et al., 2010; Volkamer et al., 2015).

**(2) Pristine terrestrial air masses:** Glyoxal mixing ratios over Patagonia and south Argentina (21 ppt) are only slightly higher than those inferred for the South Atlantic. In contrast to the other investigated terrestrial regions, in the lower and free troposphere over Patagonia glyoxal mixing ratios appear approximately constant with altitude. Mixing ratios significantly larger





than the median are only observed when approaching population centres, like Buenos Aires or Rio Grande (Argentina). In the
480 upper troposphere and distant from such emission sources of pollutants, glyoxal mixing ratios decrease further to a median of
5 ppt (Fig. 3, panel a), which is comparable to all other investigated regions. The lack of significant ground-based emission
sources and thus rather small emission of glyoxal and its precursors results in a close to constant vertical glyoxal profile over
south Argentina, in contrast to the measurements over the Amazon rainforest (Fig. 3, panel a). There, glyoxal strongly decreases
in the free and upper troposphere due to significant enhancements in the boundary layer and the lower middle troposphere that
are caused by large direct emissions of glyoxal and its precursors by biomass burning as well as secondary formation from
longer-lived precursors emitted from the rainforest (mostly isoprene; Kluge et al. (2020)). Interestingly, glyoxal in the lower
troposphere over the Amazon basin is considerably smaller (median $87 \pm 45$ ppt) than reported from the tropical rainforest in
a rural region of South-East Asia (up to 1.6 ppb) likely due to large emissions of isoprene (MacDonald et al., 2012).

**(3) Air masses affected by biomass burning:** Two of the probed biomass burning plumes (Fig. 3, brown colour) are
documented in the literature. On 12 Nov. 2019, an aged (3–4 days old) biomass burning filament was detected in the upper
troposphere and lower stratosphere over the Drake Passage around 57° S and 67° W, that most probably originated from Aus-
tralian bushfires (Kloss et al., 2021) (Fig. 3, panel d, event 3). On this day, elevated glyoxal up to $83 \pm 32$ ppt was continuously
measured during a 280 km long flight section in the upper troposphere south of Patagonia. On 8 Sept. 2019, an extended
biomass burning plume was crossed along the south Brazilian and Uruguayan coastlines towards Buenos Aires (Fig. 3, panels
a and e, event 1.1 and 1.2, and Fig. 8, panels c and d). Within this plume, the largest glyoxal mixing ratios among all biomass
burning measurements were observed (up to 3192 ppt). The FIRMS fire map as well as air mass trajectory calculations show
that the plume most probably originated from wildfires in south Brazil, Uruguay, Paraguay, and northern Argentina (see Sect.
4.2 and Fig. 8, panel c and Johansson et al. (2022)).

Based on the chemical markers, at least five more extended biomass burning plumes can be identified from the measurements
over the Tropical Atlantic (Fig. 3, events 4, 5 and 6) and over the East China Sea (event 2). Additionally, occasionally smaller
plumes were probed over the Mediterranean Sea and continental Europe during several research flights (Fig. 3, panels c and g).
Also the air masses probed over East Asia were frequently influenced by biomass burning plumes and anthropogenic emissions.
This is exemplarily indicated for event 2 (Fig. 3, panel b), which marks the crossing of polluted air masses off-coast Taipei
(Taiwan) on 19 March 2018. Around the Cape Verde Archipelago, extended biomass burning plumes were crossed 400 km
south-west of Cape Verde on 17 Aug. 2018 (event 4), later during the same flight between the islands Sal and Praia (event 5) as
well as $\sim 500$ km distant off the Brazilian coast at 1.6 km altitude on 12 Aug. 2018 (event 6). For all these encounters (except
event 4), significant enhancements in black carbon and CO were simultaneously detected.

**(4) Air masses affected by anthropogenic activities** were probed over the East China Sea, continental Europe, the English
Channel, and the Irish Sea (Fig. 3, panels b, c, f, and g, and Fig. 4, panels g–o). In these regions, large glyoxal enhancements
were observed in predominantly anthropogenic (and occasionally biomass burning) polluted air masses at all altitudes. These
most probably originated from fresh anthropogenic emissions of glyoxal and its precursor into the planetary boundary layer
(e.g. plume II with up to 547 ppt, Fig. 3, panel c), which were further transported into the upper troposphere (e.g. plume III with
up to 2970 ppt, Fig. 3, panel f). Plume III was crossed between 10.5 and 12.5 km altitude above the Malin Sea approx. 30 km



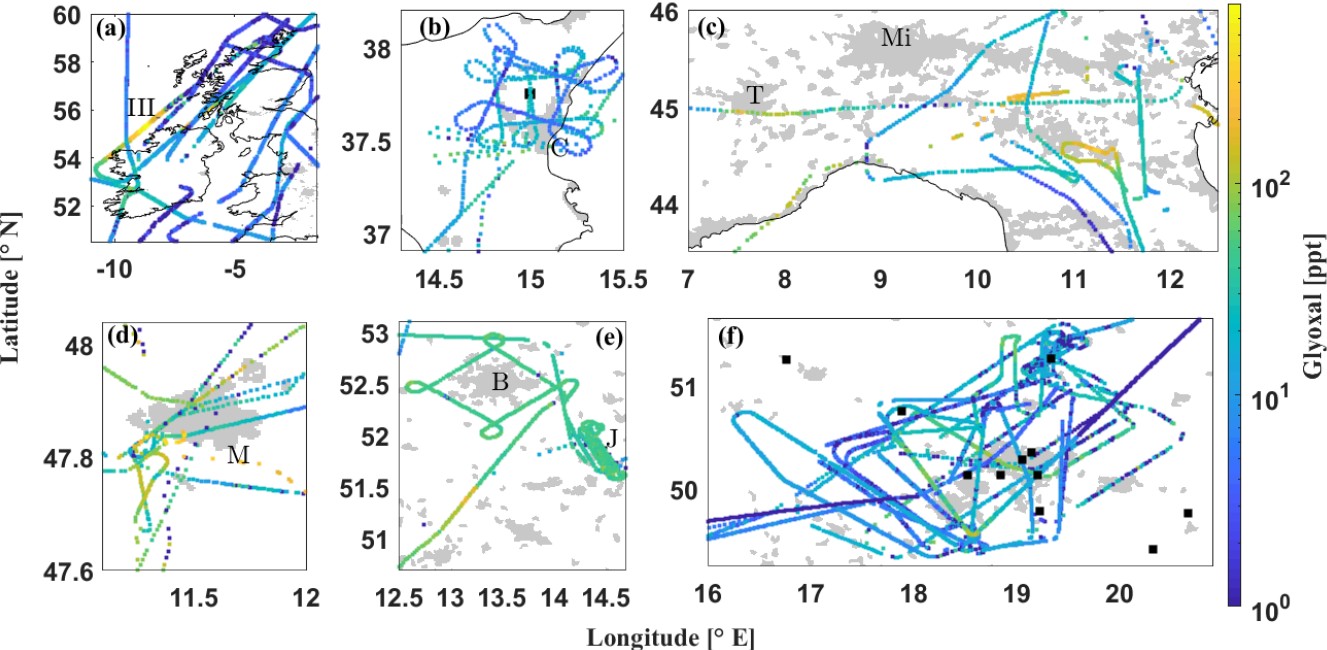

**Figure 5.** Examples of glyoxal measurements displayed over select regions of Europe: (a) over Ireland and Great Britain with the crossing of plume III during a flight leg between the Irish north coast and Scotland; (b) eastern Sicily (Italy) around Catania (C) with the Etna volcano (black square); (c) the Po Valley (Italy) with Turin (T) and Milano (Mi); (d) the Munich (M, Germany) metropolitan area; (e) the Berlin (B, Germany) metropolitan area with the Jänschwalde power plant (J) in the south-east; and (f) the upper Silesian Coal Basin (Poland) with the coal-fired power plants marked by black squares. Note the logarithmic scale of the colour axis. Grey areas mark population centres based on 2002-2003 MODIS satellite images (made with Natural Earth).

north of Ireland on 28 Sept. 2017 (Fig. 5, panel a). This plume is remarkable because of its particularly high glyoxal mixing

ratios despite the high observation altitude. Air mass trajectory calculations show that 3 to 4 days prior to the observation these air masses had been transported in a warm conveyor belt originating from the North American lower troposphere. The monitoring of a plume below 3 km altitude above the English Channel on 17 July 2017, which is one of the densest ship routes worldwide, yielded glyoxal mixing ratios up to 774 ppt (Fig. 3, panel g, plume V), which is a factor of four times less as compared to the glyoxal enhancement in the presumably much older air in plume III. Anthropogenic pollution in the remaining

but not categorized observations (Fig. 3, green) can not be excluded as the air masses are expected to be generally affected by pollution or biogenic emissions over Europe and East Asia (Andrés Hernández et al., 2022). These plumes can often not unambiguously be attributed to one of the three regimes and thus we refrain from a specific categorization.

Interestingly, the measurements in the boundary layer over continental Europe and the East China Sea yield smaller medians than those inferred over the Mediterranean Sea, where glyoxal even exceeds the observations over the Amazon rainforest

(Kluge et al., 2020) (Table 3 and Fig. 3, panels a, b, c and g).





Over continental Europe, the largest mixing ratios of 580 ppt were observed while approaching Munich airport from the west at 671 m flight altitude on 20 July 2017. Constantly enhanced glyoxal mixing ratios were observed especially over northern Italy (Fig. 3, panel c, event II, and Fig. 5, panel c), where glyoxal increased up to 522 ppt at 836 m altitude (20 July 2017). Based on all research flights in this region, we infer a median of $395 \pm 125$ ppt glyoxal in the lower troposphere in May–July 2017/2018. This is much larger than observed in the boundary layer over other European regions, e.g. Catania or Munich, with median glyoxal of 78 ppt and 51 ppt, respectively (Fig. 5, panels b and d). Glyoxal in the boundary layer over northern Italy also exceeded the observations over the Upper Silesian Coal Basin in Poland ($41 \pm 31$ ppt; Fig. 5, panel f), which is known as large emitter of anthropogenic pollutants.

Apparently, over Europe the emissions of glyoxal and its precursors from distinctive regional sources of anthropogenic pollutants are, while locally confined, potentially much stronger than those caused by wide spread biogenic and biomass burning VOC emissions, e.g. over the Amazonian rainforest (Fig. 3, panel a). Still, the median background (25–75 % of the data range) is two times larger over the Amazonian rainforest (4–65 ppt) compared to continental Europe (3–27 ppt), as expected due to the steadily larger biogenic emissions there.

The anthropogenic emissions from some larger European coastal cities (Barcelona, Marseilles and the gulf of Venice) were probed during research flights along the Mediterranean coast on 11,20, and 24 July 2017 (Fig. 3, panel g). These soundings are the reason for the possible high-biased glyoxal profile in the lower troposphere over the Mediterranean Sea, since lower altitude measurements were not performed over the remote sea, but within the plumes of coastal cities. During the measurements over the gulf of Venice on 20 July 2017, median mixing ratios of $363 \pm 118$ ppt were observed between 0.5 and 3.3 km altitude, with a maximum of $845 \pm 206$ ppt at 500 m altitude presumably caused by VOC emissions from the nearby oil refinery and the oil rigs in the Adriatic Sea and/or ship traffic (Fig. 3, panel g, plume IV). The soundings into the lower troposphere (0.5–3 km) near Barcelona on 24 July 2017 resulted in glyoxal mixing ratios up to 158 ppt (median 111 ppt). While in the lower marine troposphere of the Mediterranean Sea, glyoxal mixing ratios are even larger as over continental Europe, over the remote ocean the vertical profile decreases quickly to a few ppt at higher altitudes. This is most likely a result of the limited vertical transport by the shallow convection over the sea and the limited life-time of gloyxal and its major precursors.

The glyoxal profiles over Taiwan and the East China Sea (Fig. 3, panel b) are dominated by the polluted air masses probed during the ascent/descent into Tainan (Taiwan) airport, the low altitude soundings over Manila (Philippines), and those performed over the East China Sea between 17 March 2018 and 4 April 2018 (Fig. 3, panel b, and Fig. 4, panels m–o). The identification of the various plume origins indicate encounters of both anthropogenic emissions (e.g. plume I) as well as from biomass burning (e.g. event 2) during the flights. On 19 March 2018, while flying towards the remote East China Sea, enhanced mixing ratios of benzene and toluene indicated growing influence of anthropogenic pollutants in the probed air masses with simultaneous increase of glyoxal up to 179 ppt. Differently to the other marine measurements, lower altitude observations over the East China Sea were obtained not only close to Taiwan, but also over the more remote ocean. Given the larger emissions of pollutants by mainland China and South Korea, it is not surprising that over the East China Sea glyoxal mixing ratios in the lower troposphere significantly exceeded those above the South, Tropical and North Atlantic on average by 31 ppt (Tropical Atlantic) up to 65 ppt (South Atlantic).





Overall, the glyoxal measurements in anthropogenically polluted air masses (Mediterranean Sea, continental Europe, and East China Sea) fall into the range of previous studies in similarly polluted air masses around the globe (e.g. Lee et al. (1998); Volkamer et al. (2005a, 2007); Fu et al. (2008); Sinreich et al. (2010); Baidar et al. (2013); Kaiser et al. (2015); Volkamer et al. (2015); Chan Miller et al. (2017); Kim et al. (2022)).

### 4.2 Comparison of glyoxal VCDs between airborne and TROPOMI observations

Vertical column densities of glyoxal were detected with the Nadir observing spectrometers of the mini-DOAS instrument simultaneously to the Limb measured glyoxal concentrations. In the following, these Nadir observations and the integrated profiles are used for a detailed validation of the mini-DOAS glyoxal measurements using collocated satellite observations from the TROPOMI instrument. The potential of each observation system as well as its limitations regarding the detection of glyoxal is analysed in the different atmospheric source regimes, i.e. 1) in predominantly pristine air, where both instruments operate close to their detection limits, 2) over largely extended emission events (e.g. during the South American biomass burning season), 3) in locally confined trace gas filaments in otherwise pristine air masses (e.g. small biomass burning plumes in the marine atmosphere), and 4) in air masses generally affected by differently aged biogenic and anthropogenic VOC emissions (e.g. over industrial agglomerations of continental Europe, like the Po Valley or Upper Silesian Coal Basin).

The airborne mini-DOAS Nadir measurements are compared to respective same-day L3 processed glyoxal observations from the TROPOMI satellite instrument. Since the Sentinel-5p spacecraft was launched in October 2017, the comparison focuses on the mini-DOAS measurements from 2018 onward (EMeRGe-Asia, CoMet, CAFE-Africa (all 2018), and SouthTRAC (2019)). Additionally, the measurements over the North Atlantic (WISE, fall 2017) are included to extend the latitudinal coverage of the data set.

Quality filters already applied to the TROPOMI L3 data set (Lerot et al., 2021), the different observation geometries of the instruments, and finally the differences in the assumed a priori glyoxal profiles and the radiative transfer modelling may cause differences in the sensitivity to detect glyoxal at different altitudes j. This sensitivity is expressed by the product of the box air mass factors $B_j$ and the assumed concentration of glyoxal $[X]_j$. For the inter-comparison, the available mini-DOAS data are therefore selected according to the following criteria: 1) Due to different assumptions for $[X]_j$ and $B_j$ in the altitude layers j, the satellite and aircraft retrievals attribute different fractions of the total absorption to each j (see eq. 3). In particular, these differences of $[X]_j$ and $B_j$ yield different relative fractions $\sum_{j>i}^{j_{max}} [X]_j \cdot B_j$ of the total absorption above the aircraft flight altitude j. For a high flying aircraft, the similar observation geometries of aircraft and satellite result in a similar sum $\sum_{j_{min}}^{j_{max}} [X]_j \cdot B_j$ and hence a similar detection sensitivity for glyoxal (even though the products of single altitude layers j may be different). However, for low flight altitudes the contributions $[X]_j \cdot B_j$ are largely different. The latter primarily results from the fact, that for Nadir observations $B_j$ is larger (smaller) for altitudes j smaller (greater) than the aircraft flight altitude i as compared to those for the satellite, and less due to differences in the assumed $[X]_j$. Equally, airborne Nadir observations from low altitudes tend to be more sensitive for detecting trace gases in the lower atmosphere, than those performed from the satellite. In consequence, for an optimal and biased-minimized inter-comparison between satellite and aircraft Nadir observations (which respects the similarities in observation geometry and hence detection sensitivity), the Nadir observations from the high flying aircraft are





preferred over those performed from the low flying aircraft. In consequence of this altitude restriction, only data collected
during two research flights remain for the inter-comparison of the EMeRGe-Asia mission (Fig. 6, panel a and Fig. 9, panel b).

2) As the TROPOMI L3 data are filtered for observations with solar zenith angles $< 70°$ (Lerot et al., 2021), the same filter
is applied to the mini-DOAS data. This mostly affects the measurements over the South Atlantic as a consequence of the high
geographic latitude and consequently low sun position in the region. The largest remaining difference in the filtering of both

data sets is the TROPOMI cloud filter, which can not be applied to the mini-DOAS observations. However, the analysis of a
possible colour index dependency of the airborne Nadir glyoxal measurements (radiance at 620 nm compared to 440 nm) did
not show any correlation towards higher or lower biased observations for small (cloudy sky) or large (clear sky) colour indices.

Finally, as a consequence of the differing cruising velocities of satellite and aircraft ($\sim 7500 \, \mathrm{m \, s^{-1}}$ for the satellite and
$\sim 200 \, \mathrm{m \, s^{-1}}$ for the aircraft), for similar sized footprints (TROPOMI $3.5 \times 5.5 \, \mathrm{km^2}$ versus $(4-50) \times 0.6 \, \mathrm{km^2}$ for the mini-

DOAS measurements, depending on the signal integration time, see Sect. 2.1.3) and given that the detection limit of both
instruments is determined by the photon electron shot noise, the aircraft measurements are about a factor of six times more
sensitive than those of TROPOMI for the detection of glyoxal in individual spectra. Accordingly, to obtain the same limit for
glyoxal detection from both instruments, i.e. in order to reduce the noise in the satellite measurements, the TROPOMI glyoxal
VCDs are averaged over a $0.25° \times 0.25°$ grid. Each grid box is centred around the individual footprint of a single mini-DOAS

measurement (Fig. 7, panel b). Even though the footprints of TROPOMI and mini-DOAS geographically overlap well and only
observations from the same day are chosen, the exact timing of the measurements may differ significantly, which has to be
taken into account when comparing specific emission plumes.

The glyoxal VCDs measured by both instruments generally agree well for all investigated regions (Fig. 6), even though the
inferred VCDs show the expected statistical scatter including the occurrence of negative VCDs due to the statistical noise of the

glyoxal retrieval close to the individual detection limits. In the following, both data sets are more closely compared to highlight
their strengths and limitations for the detection of glyoxal in different situations and ranges of VCDs.

**(1) The detection of glyoxal in pristine air masses** (South Atlantic and southern Argentina) shows a good agreement among
both instruments with a corresponding scatter around zero (Fig. 6, panel d). In this region, TROPOMI and the mini-DOAS
instrument both measure slightly elevated median glyoxal VCDs of $(0.9 \pm 3.2) \cdot 10^{14} \, \mathrm{molec \, cm^{-2}}$. Compared to TROPOMI,

the mini-DOAS distribution however shows a slightly larger right side tail (Fig. 6, panel d). This is due to elevated glyoxal
detected from the aircraft (and not captured by TROPOMI) in aged biomass burning plumes from over 20 smaller wildfires
in the Chilean Biobio Region located about 100 km north-east of the HALO flight on 15 Nov. 2019. Note that the scatter
of TROPOMI data at the high resolution of $0.05° \times 0.05°$ is significantly larger than of the mini-DOAS measurements, even
though both instruments have comparable spatial resolutions. Evidently, this is a consequence of the much shorter observing

time and hence number of collected photons (about a factor of 33) from an individual pixel for TROPOMI as compared to the
mini-DOAS instrument owing to the much higher velocity of the satellite ($6.6 \, \mathrm{km \, s^{-1}}$) as compared to the aircraft ($0.2 \, \mathrm{km \, s^{-1}}$).
For daily TROPOMI data at $0.25° \times 0.25°$ resolution, this scatter is largely smoothed-out (Fig. 7, panel a), thus demonstrating
the need to spatially degrade the high resolution TROPOMI glyoxal data in glyoxal inter-comparison studies (Fig. 7, panels c





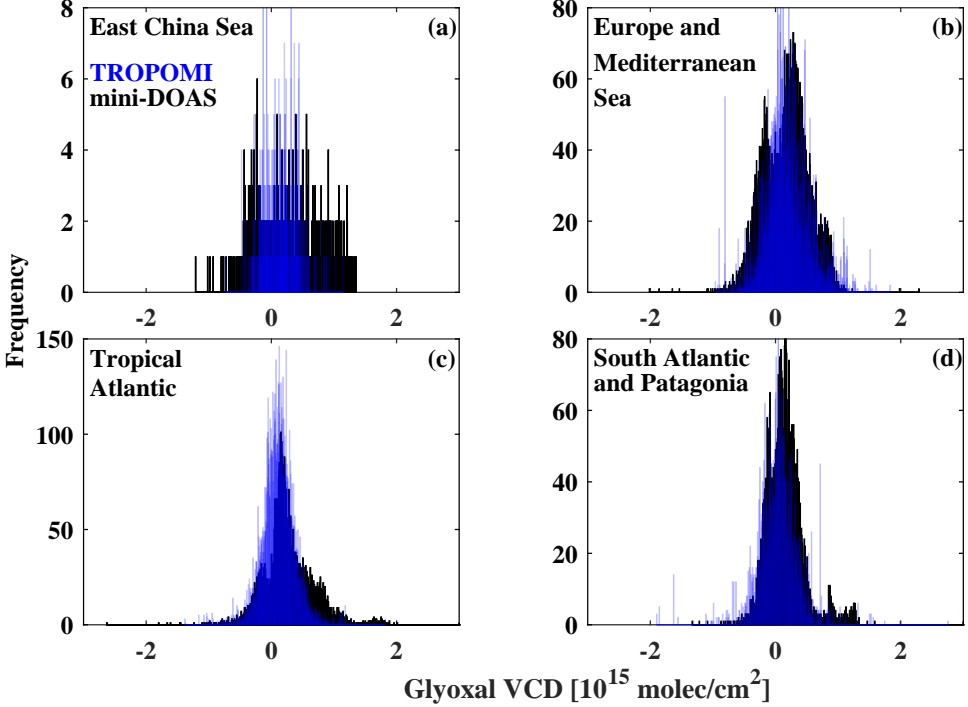

**Figure 6.** Distribution of individual glyoxal VCD measurements from the TROPOMI (blue) and mini-DOAS (black) instruments after the latter are filtered for flight altitudes $> 8\,\mathrm{km}$ and according to the TROPOMI data for solar zenith angles $< 70°$. The L3 TROPOMI data at a $0.05°$ resolution are averaged on a $0.25° \times 0.25°$ grid around the mini-DOAS measurement locations (see Fig. 7, panel b). Due to low number of airborne measurements $> 8\,\mathrm{km}$ over East Asia, the altitude filter prevents a thorough statistical analysis of the region (panel a).

and d). The respective gridding of 25 high-resolution TROPOMI measurements around each mini-DOAS observation is shown
for an example measurement in Fig. 7, panel (b).

**(2) The detection of glyoxal in largely extended biomass burning plumes** shows generally good agreement among both instruments, even though the differing overpass times and resulting plume displacements may cause some discrepancy in the along track comparison. Such large plumes were e.g. detected over the gulf of Cadiz along the southern Portuguese coast on 7 Aug. 2018 (Fig. 8, panels a and b) as well as over southern Uruguay on 7 Oct. 2019 (Fig. 8, panels c and d). Satellite images
from the MODIS instruments on the Terra and Aqua satellite taken over the gulf of Cadiz on the same day show a well confined biomass burning plume that was captured by both instruments with a respective increase in glyoxal (Fig. 8, panel a). While the TROPOMI detected glyoxal VCDs quickly decrease around the margins of the plume, the mini-DOAS instrument still measured slightly elevated glyoxal VCDs in the vicinity of the plume (Fig. 8, panel b, data below the 1:1 line). This behaviour is especially pronounced towards the open Atlantic west of the plume and might be a result of an apparent degraded sensitivity
of TROPOMI to detect small glyoxal enhancements in the lower troposphere over dark water surface, such as the ocean (see Sect. 5).





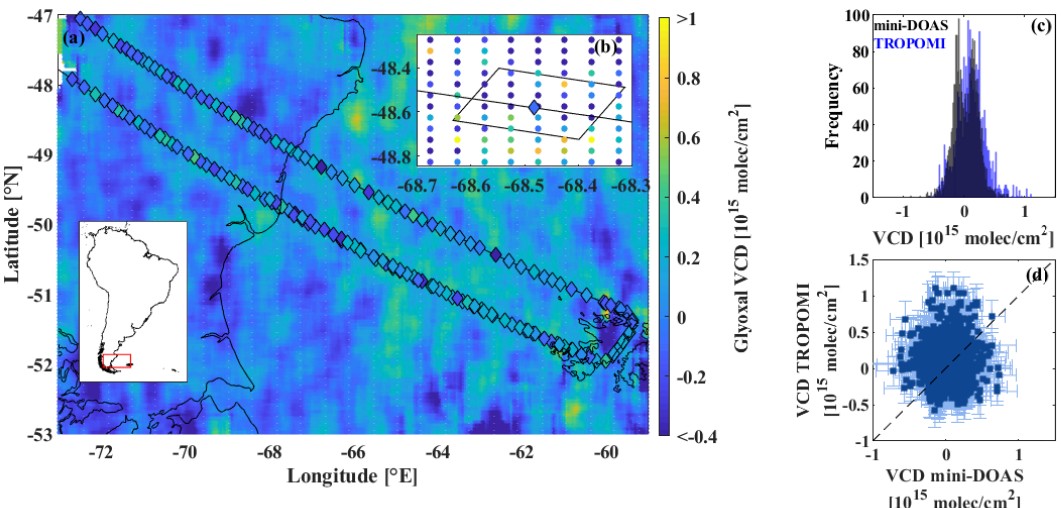

**Figure 7.** Glyoxal measurements in the pristine marine atmosphere over the South Atlantic between southern Patagonia (Argentina) and the Falkland Islands (UK) on 9 Nov. 2019. Panel (a): along track mini-DOAS glyoxal measurements (diamonds with black contours) with the collocated same-day L3 TROPOMI observations at $0.25° \times 0.25°$ resolution. For better visibility, only every fifth mini-DOAS measurement is shown. Panel (b): spatial gridding of 25 high-resolution TROPOMI measurements around an individual mini-DOAS observation. Panels (c) and (d): Frequency distribution and scatter plot of the inferred glyoxal VCDs from the mini-DOAS instrument versus TROPOMI L3 averaged on $0.25° \times 0.25°$ according to panel (b).

The glyoxal enhancements over Uruguay on 7 Oct. 2019 can not be ascribed to a specific but instead to more dispersed plumes due to extended wildfires in central Uruguay and in the north-east of Argentina. Here, the large scatter of the data in Fig. 8, panel (d) is primarily a result of the temporal mismatch between the observations of 6 h (overpass mini-DOAS around
10:45 UTC and TROPOMI at 17 UTC).

(3) **The detection of glyoxal in small or aged plumes over the oceans** with moderately enhanced glyoxal shows sizeable differences between both instruments. Such plumes predominantly originate from continental wildfires and were frequently observed in all regions during all research missions (Fig. 3). Over the oceans, such plume encounters can be well distinguished from the pristine surroundings with low glyoxal due to resulting high local concentration gradients. In contrast, for observa-
tions over land with generally higher glyoxal concentration in the background, such plume encounters with only moderately enhanced glyoxal are less distinct (Fig. 8, panels b, c, and g). During the South Atlantic and Tropical Atlantic flights, the various smaller plumes detected from the aircraft in the upper troposphere were mostly not observed by TROPOMI, e.g. along the south Chilean coast on 15 Nov. 2019 (see Sect. 4.1). During the various encounters of biomass burning plumes over the Atlantic, TROPOMI thus detects generally smaller glyoxal VCDs than the mini-DOAS instrument. This causes the right side
tail of the mini-DOAS distribution and accordingly a slight increase in the median of the VCD distributions (Fig. 6, panels c and d). The same behaviour is observed for plume encounters at lower altitudes, e.g. within the marine boundary layer below



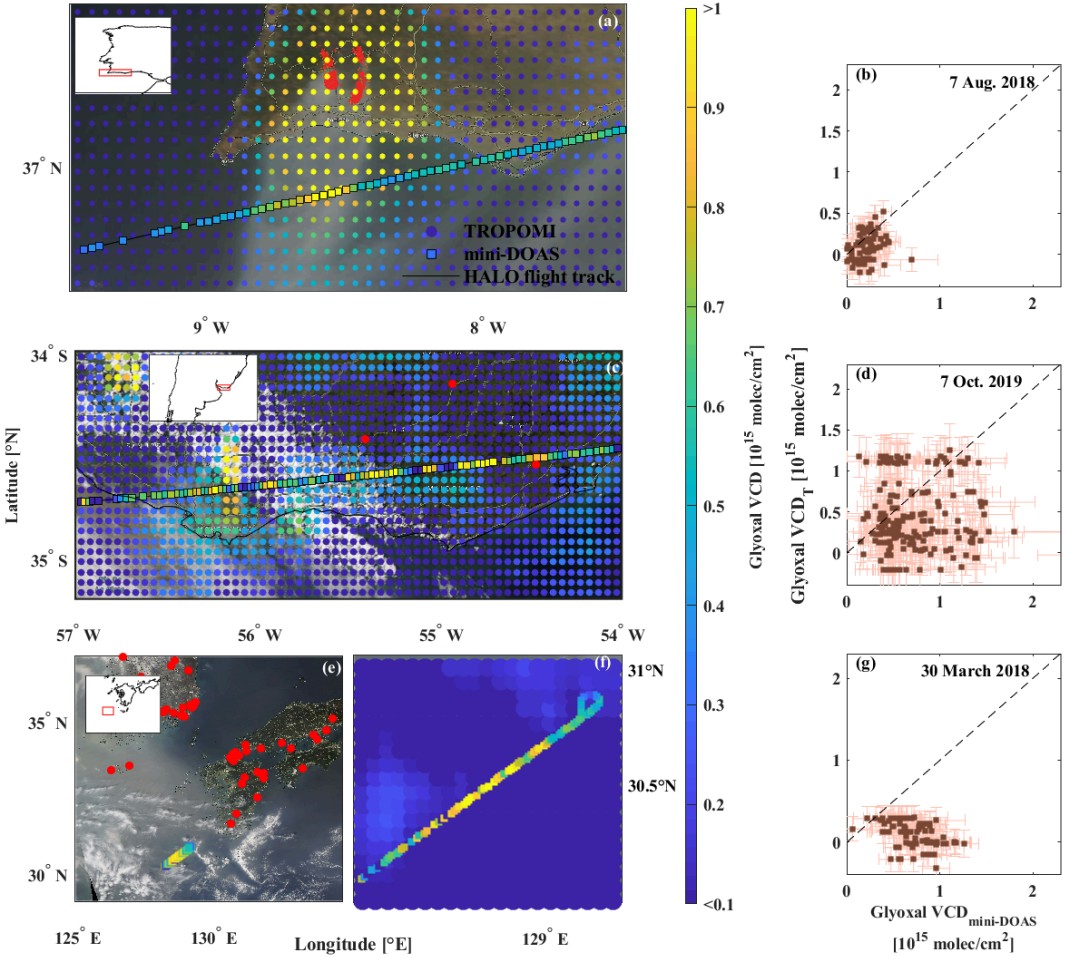

**Figure 8.** Collocated glyoxal VCDs from mini-DOAS observations (diamonds with black contours) and $0.25° \times 0.25°$ resolution L3 TROPOMI observations (squares) in different regions (panels a, c, e, and f) with corresponding scatter plots (panels b, d, and g), where each TROPOMI observation is gridded around the mini-DOAS measurements as exemplary shown in Fig. 7, panel (b). For better visibility of the underlying satellite image, the footprints are not to scale. Panels (a) and (b): Monitoring of a glyoxal plume from wildfires in the Serra de Monchique (Algarve, Portugal), detected by TROPOMI and by the mini-DOAS instrument (13 km flight altitude) over the gulf of Cadiz at 13:50 UTC and 11 UTC, respectively, on 7 Aug. 2018. Panels (c) and (d): Overpass of extended biomass burning plumes near the Uruguayan coast region observed by TROPOMI (overpass at 17:00 UTC) and mini-DOAS (8–13 km flight altitude) at 10:45 UTC on 7 Oct. 2019. Panels (e), (f), and (g): Monitoring of a glyoxal plume within the marine boundary layer above the East China Sea, south-west of Japan, on 30 March 2018 at 12:50 UTC (TROPOMI) and 1:40 UTC (mini-DOAS) with the geographic overview (panel e), zoom of the flight track (panel f), and respective scatter plot (panel g). The wildfire data (red circles, not to scale) are derived from MCD14 MODIS observations on the Terra and Aqua satellites. The MODIS satellite images are taken from NASA WORLDVIEW based on the satellite overpasses on 13:48 UTC on 7 Aug. 2018 (panel a), 17:23 UTC on 7 Oct. 2019 (panel c), and 2:30 UTC on 30 March 2018 (panel e).



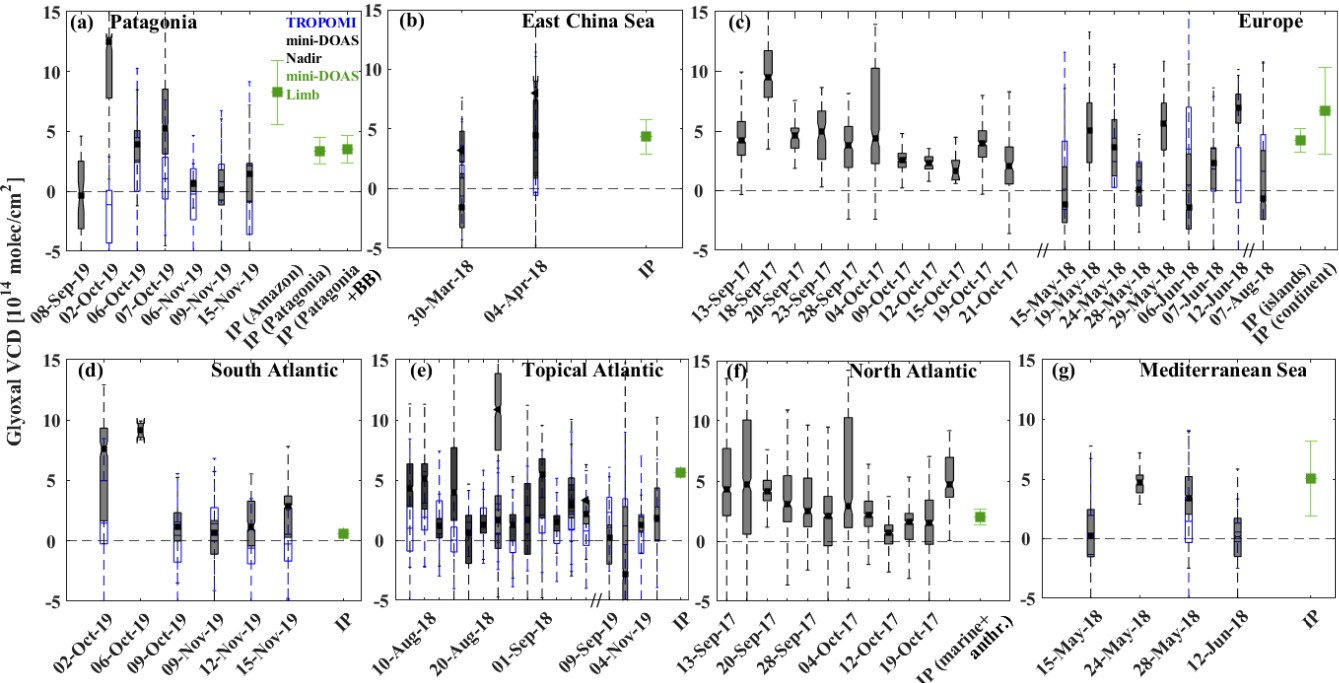

**Figure 9.** Comparison of inferred glyoxal VCDs for TROPOMI (blue) and mini-DOAS measurements (black) for the different regions: South American continent (panel a), East China Sea (panel b), continental Europe and its northern islands (panel c), South Atlantic (panel d), Tropical Atlantic (panel e), North Atlantic (panel f), and the Mediterranean Sea (panel g). The whisker boxes indicate the 25th to 75th percentile ranges and respective medians with in-range minima and maxima. For comparison, also the integrated glyoxal Limb profiles (IP, see Sect. 4.1) for the different subsets of data are shown (green squares). For the South American Continent, mini-DOAS VCD measurements are only available for the Patagonia and Argentina SouthTRAC deployments in fall 2019. For the measurements before 2018, TROPOMI data were not yet available.

2 km over the East China Sea south of Kyushu island (Japan) on 30 March 2018 (Fig 8, panels e, f, and g), or at 1.6 km altitude and ∼ 500 km north-east of the Brazilian coast over the Tropical Atlantic on 15 Aug. 2018 (see Sect. 4.1), which were both not detected by TROPOMI. Similar observations were made during multiple other occasions, e.g. during biomass burning

plume encounters near the Canary Islands on 7 and 15 Aug. 2018 and 7 Sept. 2018, off-coast Brazil on 4 Nov. 2019, along the West African coast on 10 and 31 Aug. and 4 Sept. 2018, and along the coast of Japan east of Osaka on 4 April 2018. Very likely, the undetected increase of glyoxal in such small or low lying biomass burning plumes by TROPOMI is a result of (a) the potentially small vertical extent of some of the low lying plumes and hence their relatively small contribution to the total atmospheric VCD, (b) the small horizontal extent of the plumes, which may just not be detectable by the satellite due to the

lower spatial resolution of TROPOMI, or simply missed due to the temporal mismatch with the airborne observations, (c) an incorrect a priori glyoxal profile in the air mass factor calculations, or (d) a decreased sensitivity of the TROPOMI instrument to low lying plumes over surfaces with low reflectivity (e.g. oceans), or a combination of all these circumstances.





**(4) The detection of glyoxal in mixed polluted continental or coastal air masses** (e.g. over continental Europe) agrees well between both instruments for upper tropospheric measurements, even though glyoxal VCDs are sometimes underestimated

by TROPOMI (e.g. over the Upper Silesian Coal Basin). While we generally restrict the comparison to aircraft observations in the upper troposphere, we additionally found the airborne instrument to generally measure more glyoxal than the satellite for measurements closely above and within the polluted boundary layer. This is primarily a consequence of the observation geometry, the different a priori assumptions, and the proximity of the measurements to the pollution. At lower flight altitudes, a relatively larger fraction of photons detected by the aircraft instrument will have travelled through the pollution layer than the

photons detected by the satellite. Secondly, the mostly larger assumed a priori glyoxal for the airborne observations as compared to MARGRITTE CTM simulations for low altitudes in polluted environments leads to a larger detection sensitivity for glyoxal of the aircraft at these altitudes. This becomes in particular noticeable when the aircraft is close to within the polluted lower atmosphere. For such measurements, the box air mass factors give even more weight to the glyoxal located below the aircraft than to the glyoxal fraction above the aircraft (to which the satellite relatively gives more weight). Over continental Europe,

both instruments detected median glyoxal VCDs roughly two to five times larger than over pristine marine environments, such as over the South, Tropical, and North Atlantic (Fig. 9, panels c, d, e, f, and g). When probing distinct (mostly anthropogenic) emission sources of glyoxal and its precursors such as over major population centres (e.g. Bologna) or industrial agglomerations (e.g. the Po Valley or the upper Silesian Coal Basin), TROPOMI generally measures smaller VCDs than the mini-DOAS instrument (right side tail in the mini-DOAS distribution in Fig. 6, panel b). Accordingly, over the Upper Silesian Coal basin

a factor of five times larger glyoxal VCDs are detected by the mini-DOAS instrument ($(7.2 \pm 5.7) \cdot 10^{14}$ molec cm$^{-2}$) than by TROPOMI ($(1.3 \pm 3.7) \cdot 10^{14}$ molec cm$^{-2}$). The smaller VCDs reported by TROPOMI over populated and industrial areas with a correspondingly increased aerosol load in the boundary layer might be caused by the decreased detectability of glyoxal from space. Figure 9 shows the glyoxal VCDs inferred for collocated flight sections in the different regions and seasons with the median, 25th and 75th percentiles (box edges) and whiskers indicating in-range minima and maxima. Additionally, the

glyoxal VCDs inferred from the integrated Limb profiles are compared to the Nadir inferred VCDs (Fig. 9, green squares). The VCDs obtained from the integrated vertical glyoxal profiles (IP) are in good agreement to the Nadir measured VCDs, thus providing confidence in the consistency of the airborne glyoxal measurements. In general, a good agreement is found between the airborne and space-borne glyoxal measurements for all regions and seasons, with the exceptions discussed above.

Even though (aged) biomass burning plumes often only extend over a limited altitude range in the lower and middle tropo-

sphere and therefore may only contribute a minor fraction to the total VCDs in background air, they are apparently discernible in airborne observations and in cases of more pronounced pollution or large vertical extent even from space (e.g. Fig. 8, panels a and b, and Fig. 9, panels a and c).

### 4.3 Comparison of measured airborne and EMAC simulated glyoxal

In the following, the airborne glyoxal measurements are compared to EMAC model simulations. The simulations of glyoxal

concentrations and VCDs are performed on a 10 min time grid resolution along the flight trajectories based on the settings described in Sect. 3.3. For the measured versus modelled inter-comparison, VCDs detected at all flight altitudes are included in





the analysis. This comes with the advantage of a larger number of available VCD measurements than for the airborne to satellite comparison (in particular over the East China Sea), but at the same time the flight medians slightly change from those compiled for the satellite comparison (Fig. 9 and Fig. 11). For the comparison, we distinguish between observations in background air (outside of identified emission events) and observations of elevated glyoxal due to specific emission plumes (see Fig. 3 and Sect. 4.1). The resulting profiles differentiate the characteristic background glyoxal in each region (green, yellow, and blue in Fig. 10) from local glyoxal enhancements due to biomass burning or anthropogenic pollution (brown and black colours, respectively, in Fig. 10). Since for low glyoxal concentrations the measurement noise occasionally leads to negative data (see Fig. 6 and Fig. 11), and the model does not reproduce such measurement noise, inferred negative glyoxal observations are omitted from the comparison.

For the employed EMAC set-up, simulated glyoxal underestimates measured glyoxal to varying degrees, in agreement with previous findings (e.g. Myriokefalitakis et al. (2008); Fu et al. (2008); Stavrakou et al. (2009a); Walker et al. (2022); and Figs. 10 and 11). The best agreement between the model and measurement is generally found for the upper troposphere and in the most pristine regions with only little surface emissions or expected long-range transport of glyoxal and its precursors, i.e. over Patagonia or the North Atlantic (Fig. 10, panels a, e, and g, and Fig. 11, panels a, d, and f). Larger differences of modelled and measured glyoxal are found for regions with significant biogenic (e.g. tropical rainforests), anthropogenic (e.g. continental Europe, East China or Mediterranean Sea; Fig. 10, panels c, d, and h, and Fig. 11, panels b, c, and g), or biomass burning related emissions of glyoxal and precursor VOCs (e.g. the Tropical Atlantic, Fig. 10, panel f, and Fig. 11, panel e). In the mixed polluted background atmosphere over continental Europe, the comparison shows a relatively small glyoxal underestimation by EMAC (Fig. 10, panel d, green), whereas measured and modelled glyoxal differ significantly when probing local emission hotspots (e.g. city plumes; Fig. 10, panel d, grey) as well as in the mixed polluted marine boundary layer over both the East China and Mediterranean Sea (Fig. 10, panels c, h).

The following three key findings are eminent:

**(1)** Over the tropical rainforest, with its significant surface emissions of biogenic VOCs, the model overestimates glyoxal by a factor of 2–3 in the planetary boundary layer and free troposphere (Fig. 10, panel b). At higher altitudes (>8 km), measured glyoxal exceeds the simulations, as also observed for the other investigated regions. This may indicate too strong emissions of short-lived biogenic VOCs (e.g. isoprene) from the rainforest and/or underestimated emissions of longer-lived glyoxal precursor molecules (e.g. aromatics, aliphatic compounds) within the model. In addition, the strong overestimation might point to missing glyoxal loss processes like the uptake and oxidation in cloud droplets which have been recently represented in EMAC in the Jülich Aqueous-phase Mechanism of Organic Chemistry (JAMOC; Rosanka et al., 2021b, a).

**(2)** In distinctive biomass burning or anthropogenic emission plumes (Fig. 10, brown and grey), the glyoxal underestimation from the model is found to be larger than outside of the plumes. For example, glyoxal enhancements due to city plumes over Europe are mostly reproduced by the model, however with the magnitude underestimated on average by a factor of 8. Some of the underestimation of these specific plume events is related to the fact that EMAC does not represent any monthly variation in the anthropogenic VOC emissions and does not represent any daily temporal evolution of specific biomass burning events. At the same time, the coarse resolution used (about $209 \times 209$ km$^2$) by EMAC smooths out specific plumes observed by the





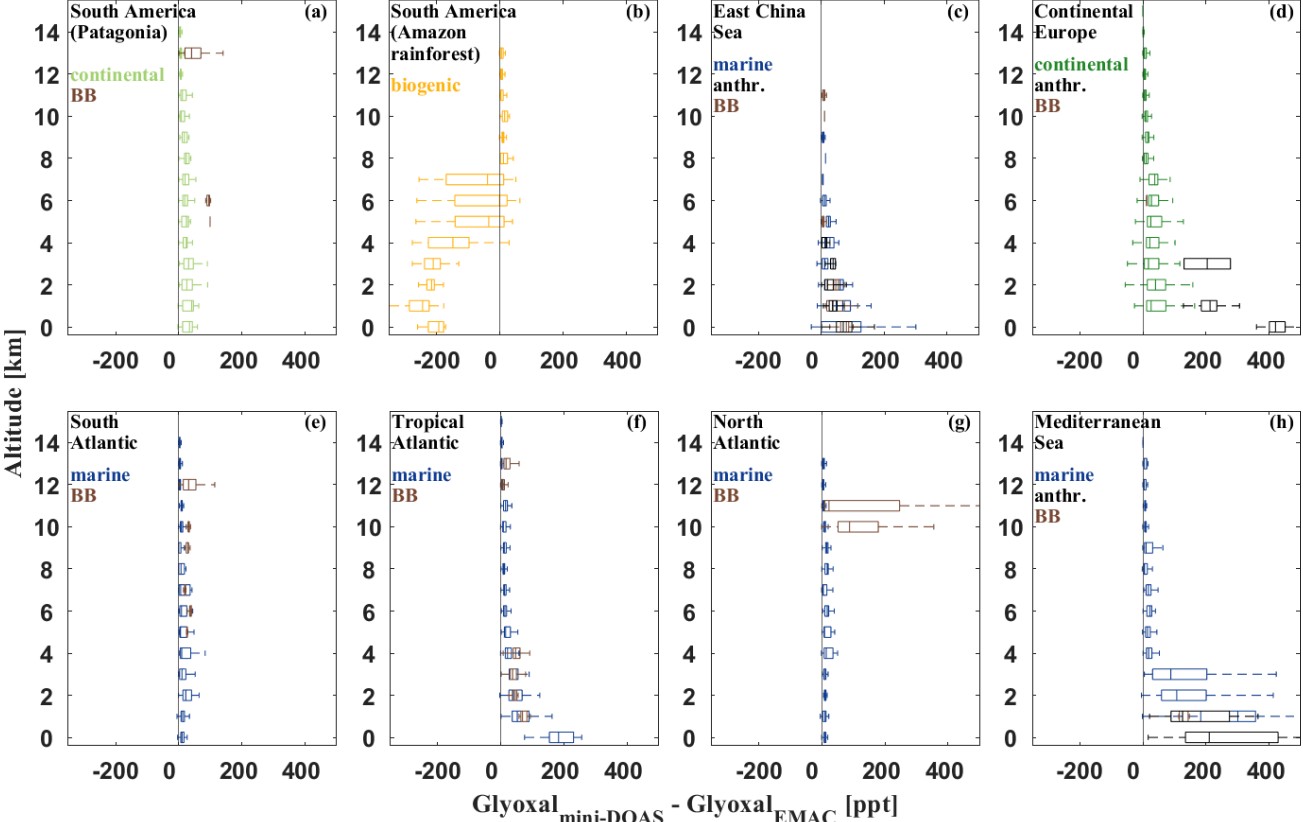

**Figure 10.** Difference of measured airborne and EMAC simulated glyoxal for the different regions (continental measurements are shown in green, over the rainforest in yellow, and marine observations in blue). For the comparison, only positive valued measurements are considered due to the missing randomness of the model simulations (see text). Since for the measurements over the Amazon rainforest in 2014 and over the Mediterranean and Near East in 2015 no EMAC simulations are available, simulations performed for the year 2017 are used instead (panel b and a minor part of the data in panel h). The data are plotted in $1\,\mathrm{km}$ altitude bins for the 25th to 75th percentiles (whisker boxes) with respective medians and with in-range minima and maxima. Profiles for identified emission events (see Fig. 3 and Sect. 4.1) are calculated separately (biomass burning in brown and anthropogenic emissions in black).

HALO observations. Further reasons for the underestimated glyoxal in polluted air masses are presently unclear, but may be related to incorrect assumptions regarding the strengths and composition of the anthropogenically emitted cocktail of glyoxal producing VOCs.

**(3)** Our findings of enhanced glyoxal in the tropical marine boundary layer are in agreement with previous reports from the Tropical Pacific and Atlantic (e.g. Sinreich et al. (2010), Walker et al. (2022)), but are not reproduced by the model (Fig. 10, panel f). Over the Tropical Atlantic, EMAC simulates an approximately constant vertical profile with much smaller median glyoxal mixing ratios ($4\,\mathrm{ppt}$) than observed ($\sim 44\,\mathrm{ppt}$). In the free and upper troposphere, the simulations and observations





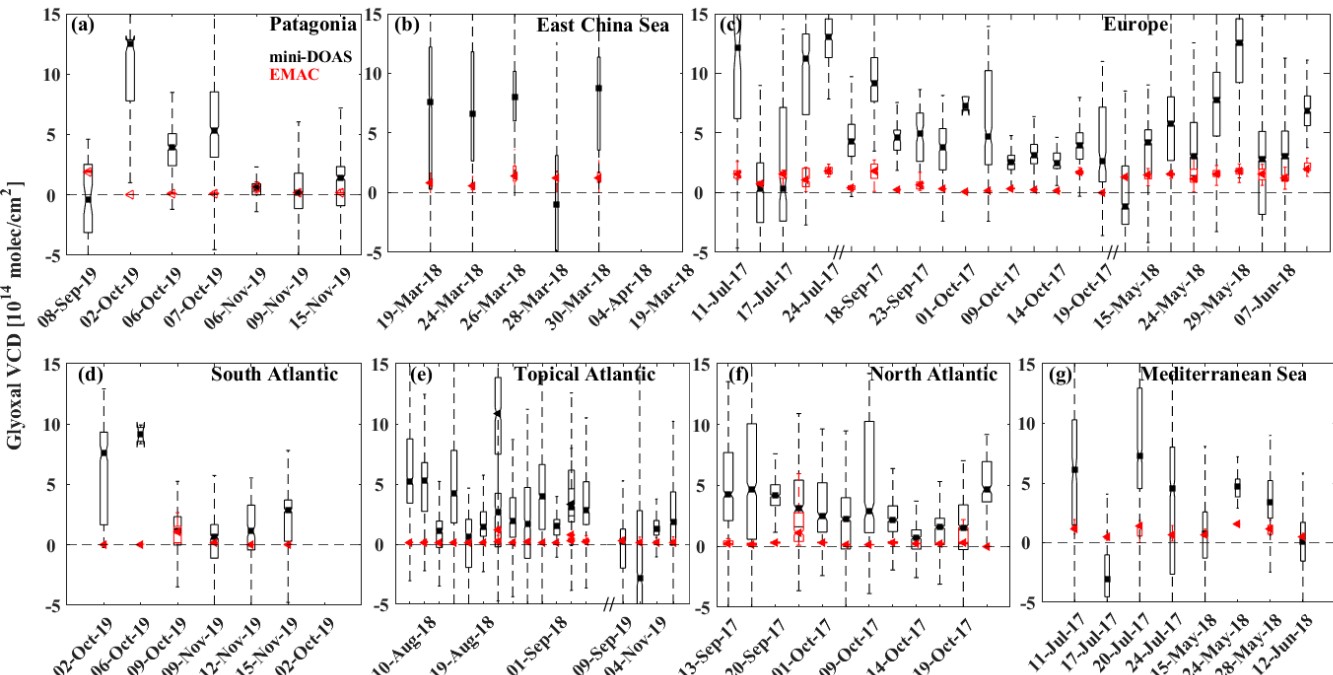

**Figure 11.** Comparison of measured airborne (black) and simulated (red) glyoxal VCDs in the different regions and for all flight altitudes of the measured data.

outside of biomass burning plumes agree better. Again, this discrepancy may point to missing glyoxal sources related to the organic micro-layer at the sea surface in the model (e.g. Chiu et al. (2017); Brüggemann et al. (2018)).

## 5  Discussion

As discussed in Sect. 4.2, the airborne mini-DOAS and satellite TROPOMI observations of glyoxal overall show a good agreement for measurements in (a) pristine air with low glyoxal concentrations near the detection limits of the instruments, (b) polluted air masses with high glyoxal concentrations (e.g. major emission plumes), and (c) continental air of mixed pollution sources. Exceptions to this overall good agreement are observations (a) over land with an increased aerosol load in the boundary layer, (b) of low lying plumes (mostly biomass burning) over the ocean with its low reflectivity in the visible wavelengths, and (c) of pollution plumes over the oceans with a limited vertical or horizontal extent irrespective of their altitude. For TROPOMI, this also leads to underestimated glyoxal in the marine surroundings of larger emission plumes.

A more systematic comparison of the mini-DOAS glyoxal measurements to previous air- or ship-borne measurements in the same regions, seasons, and altitudes is complicated by the limited number of respective studies in the remote marine free and upper troposphere. In this respect, our observations provide a unique and novel data set of glyoxal measurements covering a large number of so far little explored (or even unexplored) regions and altitude ranges with different pollution levels around the





globe. In particular, the data may provide new insights into the vertical profiles of glyoxal in different pristine marine regions. Moreover, they provide further evidence of elevated glyoxal in aged and long-range transported biomass burning plumes,

similar to past studies that reported much larger glyoxal concentrations than expected. These two aspects of our measurements are discussed in more detail in the following.

(1) Our measurements confirm previous reports of (at least) occasionally elevated glyoxal (1–140 ppt) in the marine boundary layer over the remote ocean (e.g. Zhou and Mopper (1990); Sinreich et al. (2010); Coburn et al. (2014); Mahajan et al. (2014); Lawson et al. (2015); Volkamer et al. (2015); Chiu et al. (2017); Walker et al. (2022)). The measurements further con-

firm observations of elevated glyoxal in the marine boundary layer over tropical oceans as compared to the marine boundary layer at higher latitudes. In fact, we observe 2–4 times more glyoxal in the marine boundary layer of the tropics as compared to the South or North Atlantic (Fig. 3 and Table 3). The comparable large range of observed glyoxal mixing ratios in different marine regions is indicative for the large variety of the types and transport history of the investigated air masses (Table 3). Yet, given the short lifetime of glyoxal in the sunlit atmosphere of $\sim 2\,\mathrm{h}$ (Koch and Moortgat, 1998; Volkamer et al., 2005a; Tadić

et al., 2006; Fu et al., 2008; Wennberg et al., 2018), the observation of elevated glyoxal in the tropical marine atmosphere requires rather large and potentially variable sources in the range $0.5\,\mathrm{ppt\,h^{-1}}$ to $70\,\mathrm{ppt\,h^{-1}}$, if glyoxal is derived from the photochemical decay of its organic precursor substances. In the past, two major explanations for the yet unexplained glyoxal over marine regions have been discussed, i.e. either (a) the transport of longer-lived glyoxal precursor species from land-based sources into the remote marine atmosphere, and/or (b) the emissions of glyoxal and/or its organic precursors from the ocean.

(a) Based on co-located measurements of glyoxal and some of its main known precursors like isoprene and the monoterpenes, Lawson et al. (2015) estimated their potential contribution to glyoxal in the marine boundary layer to be on the order of 10%. Additional long-lived glyoxal precursor molecules (e.g. aromatics, acetylene, or larger VOCs) have been suggested to explain the discrepancy between observed and expected glyoxal in the pristine marine boundary layer, but conclusive answers have not yet been found (e.g. Sinreich et al. (2010); Rinaldi et al. (2011); Coburn et al. (2014); Mahajan et al. (2014); Lawson et al.

(2015); Chiu et al. (2017); Walker et al. (2022)).

(b) Direct emissions of significant amounts of glyoxal from the oceans could also be convincingly ruled out, primarily because of its high water solubility (effective Henry's law coefficient $H^{cp} = 4100\,\mathrm{mol\,m^{-3}\,P^{-1}}$, Sander (2015)) and the observation that the daytime flux of glyoxal is directed from the atmosphere into the ocean (Zhou and Mopper, 1990; Coburn et al., 2014; Chiu et al., 2017; Zhu and Kieber, 2019). Potentially relevant glyoxal precursor molecules include glycolaldehyde

and acetaldehyde. However, their contribution to the atmospheric glyoxal budget is poorly constrained. For instance, acetaldehyde is largely produced in seawater (Zhu and Kieber, 2020) and a net flux to the atmosphere is expected (Zhu and Kieber, 2019). Estimates of the global oceanic source of acetaldehyde range from 34 to 57 $\mathrm{Tg\,yr^{-1}}$ (Millet et al., 2010; Wang et al., 2019). Usually global atmospheric models neglect this source, as is the case in the EMAC simulation performed in this study. However, even when this acetaldehyde source is taken into account atmospheric models still underestimate observations in the

boundary layer and free troposphere (Wang et al., 2019). The implied significant and widespread missing source of acetaldehyde may therefore be relevant for the global glyoxal budget. Direct oceanic emissions of unsaturated aliphatic or additional aromatic glyoxal precursor species besides acetylene (e.g. benzene, toluene, ethylbenzene, or xylenes) have also been found


to be insufficient to explain the observed glyoxal concentrations in the marine environment of the tropics (Xiao et al., 2007; Mahajan et al., 2014), even though their potential source strengths have recently been reported to be larger than previously

thought (Rocco et al., 2021). At night, $NO_3$ may oxidise some organic VOCs (e.g. toluene) and thus build-up a certain glyoxal level until dawn, but the potential production rate is far too small to explain the observed glyoxal concentrations both at night and during daytime (Coburn et al., 2014; Walker et al., 2022).

In contrast, recent laboratory experiments have shown that UV light-initiated reactions at the sea surface organic microlayer involving DOC (dissolved organic carbon) may lead to the production of significant amounts of VOCs of low solubility,

e.g. fatty (heptanoic, octanoic) and nonanoic (NA) acids, and thus of secondarily formed oxidised VOCs like glyoxal and its precursors (Ciuraru et al., 2015; Chiu et al., 2017). From their study, Chiu et al. (2017) concluded that the ozonolysis of 2-nonenal is most likely the primary chemical mechanism to produce significant amounts glyoxal in the marine atmosphere, and that this source can potentially sustain tens of ppt glyoxal over the ocean. In addition, a recent study of Brüggemann et al. (2018) discusses an abiotic source of organic vapours emitted by photochemical reactions of the amphiphilic compounds

forming surfactants at the sea surface. In their study, they determined global emissions to be 23.2–91.9 $\mathrm{TgC\,yr^{-1}}$ of these organic vapours due to interfacial photochemistry, of which 1.11 (0.70–1.52) $\mathrm{Tg\,yr^{-1}}$ are attributed to emissions of isoprene. Though potentially relevant for organic aerosol mass over the remote ocean (Brüggemann et al., 2018), at this point it is unclear how much this organic vapour may ultimately contribute to the elevated glyoxal observed over the tropical oceans.

In conclusion, the low-to-high latitudinal gradient of glyoxal and the comparable low concentrations in the free troposphere

also above the biologically active tropics, provide some evidence that indeed the emissions of DOC related VOCs from the oceanic micro-layer and their photochemical decay rather than long-range transport of long-lived glyoxal precursors from land is primarily responsible for the elevated glyoxal in the pristine marine atmosphere of the tropics. A more detailed discussion of potentially relevant glyoxal precursor molecules suitable to explain the observations of elevated glyoxal in the marine (and in particular tropical) environment and how they are represented in the global circulation chemical model EMAC is beyond the

scope of the present study, but will be subject of an accompanying study by Rosanka et al., (2022) (manuscript in preparation, 2022).

**(2)** Further, our observations of enhanced glyoxal in aged biomass burning plumes both over land and the ocean, confirm recent reports of enhanced glyoxal in aged biomass burning plumes that have been transported for at least several days (e.g. Alvarado et al. (2020)). Such plumes were observed on multiple occasions over the South (one event), North (one event), and

Tropical Atlantic (multiple events, see Sect. 4.1) within the framework of the present study. Again, due to the short atmospheric lifetime of glyoxal during daytime, the glyoxal detected in these aged biomass burning plumes was necessarily secondarily formed from yet unidentified longer-lived VOC precursor species (e.g. benzene, acetylene, or aromatics), which were co-emitted during the wildfires. Details on how the primary emitted and secondary formed glyoxal producing VOCs evolve in aged biomass burning plumes still need to be explored before more firm conclusions on the fate of glyoxal and its potential to

form secondary aerosols in these aged biomass burning plumes can be drawn. Nevertheless, significant amounts of glyoxal may be produced by oxidation of aromatic compounds from evaporation of organic aerosols in such air masses (Palm et al., 2020). Accounting for this process could also partially resolve the model underestimation of ozone production by biomass burning





emissions (Bourgeois et al., 2021). Since biomass burning is much more frequent in the tropics than at higher latitudes, the glyoxal formed in these aged biomass burning plumes from long-lived precursors may also enhance the observed low to high

latitude gradient of glyoxal in the lower marine atmosphere.

Our study confirms recent findings on glyoxal in the atmosphere, but it also offers new aspects on how wide spread elevated glyoxal occurs in the atmosphere. This emphasizes the potential role glyoxal may play in the oxidation of VOCs, the oxidative capacity of the atmosphere and hence ozone formation, and on its importance in secondary aerosol formation.

## 6    Conclusions

We report on spectroscopic glyoxal measurements in Nadir and Limb geometry performed during 72 research flights all around the globe from aboard the German research aircraft HALO between 2014 and 2019. The directly measured and profile integrated column densities of glyoxal are compared to near collocated measurements from space by the TROPOMI instrument on the Copernicus Sentinel-5 Precursor satellite. Based on this unique data set with respect to geographical and seasonal coverage, an in-depth evaluation of the strengths and weaknesses of each observation technique is made. Overall a good agreement is

found among the two data sets, with the exception of airborne observations of faint glyoxal plumes occurring over surfaces of low reflectivity and plumes in lower altitudes (i.e. in the marine or planetary boundary layer).

The combined and validated airborne Nadir and Limb data set is the first of its kind and may offer new information into the fate of atmospheric glyoxal in the global atmosphere. The Limb measurements further allow us to infer and investigate different glyoxal profiles in the troposphere over different regions of the world. The integrals of these Limb profiles compare well with

the Nadir total glyoxal column densities measured from the aircraft and from space. Both types of airborne measurements, i.e. total atmospheric column densities and the vertical profiles, are further compared to glyoxal simulations of the global atmosphere-chemistry model EMAC. The comparison of measured and simulated glyoxal point to several deficits in the current representation of the photochemistry of glyoxal and its precursor species in respective models. The general underestimation of glyoxal found in the simulations over land and oceans has already been recognized in previous studies, most of which were

based on past satellite observations (e.g. Fu et al. (2008); Myriokefalitakis et al. (2008); Stavrakou et al. (2009a)).

Our airborne glyoxal observations confirm key findings related to atmospheric glyoxal reported in recent studies, specifically the occurrence of elevated glyoxal in aged biomass burning plumes (Alvarado et al., 2020) and in the marine boundary layer of the tropics (Sinreich et al., 2010; Rinaldi et al., 2011; Coburn et al., 2014; Mahajan et al., 2014; Lawson et al., 2015; Chiu et al., 2017; Walker et al., 2022). In addition, our measurements provide novel insights into various aspects of atmospheric glyoxal,

e.g. its height distribution in rarely or yet unprobed air mass types and/or those elusive for glyoxal detection from space.

Moreover, the study points to some major deficits in our current understanding of atmospheric glyoxal. When combined, these deficits reveal multiple causes for the current glyoxal underestimation, which are not resulting from the disregard of a single glyoxal precursor molecule, source, or single chemical pathway, but potentially from a suite of glyoxal precursor molecules and formation processes. This conjecture is supported by the observed deficits in explaining the measured glyoxal

in different types of air masses, i.e. in (a) anthropogenic plumes of larger agglomerations, (b) aged polluted air masses forming





the continental glyoxal background, (c) pristine air masses of the marine boundary layer in the tropics, (d) the pristine marine atmosphere (e.g. South and North Atlantic), and (e) aged biomass burning plumes. Our observations provide novel information on the required emission strengths, concentration and lifetimes of the possible different glyoxal producing precursors and their intermediates necessary to close the apparent observation to model gap.

In this respect, it is noteworthy to acknowledge that deficits in understanding atmospheric glyoxal ultimately indicate a more fundamental deficiency in the current knowledge of the photochemistry and emissions of VOCs in the atmosphere, with a variety of consequences for the oxidative capacity and ozone in the atmosphere. Moreover, since glyoxal is known to support secondary organic aerosol (SOA) formation, our finding of overall larger glyoxal in both the polluted and pristine atmosphere provides evidence for an overall larger role glyoxal may play in global SOA formation. Our study as well as previous research

on atmospheric glyoxal thus strongly motivates a reevaluation of the current understanding of global VOC chemistry and its implications, e.g. for the oxidative capacity of the atmosphere, the formation of ozone and of secondary formed aerosols, both of which impact human health, the atmospheric radiative balance, and hence the global climate.

*Data availability.* The mini-DOAS data are archived in the HALO data depository (https://doi.org/10.17616/R39Q0T, re3data.org (2022)) and can be accessed upon signing a data protocol. Access to TROPOMI glyoxal tropospheric column data is possible via the GLYRETRO

website (https://doi.org/10.18758/71021069, Lerot et al. (2021)). The EMAC simulation data are archived at the Jülich Supercomputing Centre (JSC) and are available upon request.

*Author contributions.* FK, TH, KP, MR, and BW operated the mini-DOAS instrument. CL developed the TROPOMI glyoxal product. SR performed the EMAC simulations. CL, KP, SR, and DT contributed to the interpretation of the data analysis. FK performed the data analysis and wrote the manuscript with contributions from KP, CL, SR, and DT.

*Competing interests.* The authors declare that they have no conflict of interest.

*Financial support.* The funding of the HALO aircraft as well as the contributions to the various missions via the German Research Foundation (DFG, HALO-SPP 1294), the Max-Planck Society (MPI), the Helmholtz-Gemeinschaft and the Deutsches Zentrum für Luft- und Raumfahrt (DLR) (all from Germany) are highly acknowledged. The scientific work of FK, KP, MR, and BW was supported via the German Research Foundation (DFG) through grants PF-384/7-1, PF384/9-1, PF-384/16-1, PF-384/17, and PF-384/19. The S5p/TROPOMI glyoxal

product has been supported by the European Space Agency via the GLYRETRO project, part of the Sentinel-5p+Innovation programme (contract no. 4000127610/19/INS). The authors gratefully acknowledge the Earth System Modelling Project (ESM) for funding this work by providing computing time on the ESM partition of the supercomputer JUWELS at the Jülich Supercomputing Centre.





*Acknowledgements.* This work contains modified Copernicus Sentinel-5 Precursor satellite data (2018–2019). We acknowledge the use of imagery from the NASA Worldview application (https://worldview.earthdata.nasa.gov/, last access: 19 May 2022), part of the NASA Earth
Observing System Data and Information System (EOSDIS). We thank the Deutsches Zentrum für Luft- und Raumfahrt (DLR) for their support during the certification process of the mini-DOAS instrument. Special thanks are given to Lisa Kaser, Martina Hierle, Frank Probst, Andreas Minikin, Andrea Hausold, Michael Großrubatscher, Stefan Grillenbeck, Marc Puskeiler for flight coordination and planning, to Alexander Wolf and Thomas Leder, the flight engineers, and to the BAHAMAS team. We are grateful to our colleagues for their support and coordination of the different research missions, namely Manfred Wendisch (University of Leipzig, Germany), Ulrich Pöschl and Meinrat
Andreae (both Max-Planck Institut for Chemistry, Mainz, Germany) for the ACRDICON-CHUVA mission, Hartwig Harder and Jos Lelieveld (both with Max-Planck Institut for Chemistry, Mainz, Germany) for OMO and CAFE-Africa, Maria Dolores Andrés Hernández and John Burrows (both with the University of Bremen, Germany) for EMeRGe-EU and EMeRGe-Asia, Peter Hoor (University of Mainz, Germany) and Martin Riese (Forschungszentrum Jülich, Germany) for WISE, Andreas Fix (DLR, Germany) for CoMet, and finally Martin Riese (Forschungszentrum Jülich, Germany), Peter Hoor (University of Mainz, Germany) and Markus Rapp (DLR, Germany) for SouthTRAC. We
are grateful to Ulrich Schumann (DLR, Germany) for compiling and evaluating the measured data from the HALO-FAAM formation flight over south Germany on 13 July 2017 (https://doi.org/10.5281/zenodo.4427965). We are grateful to Eric Förster, Peter Hoor, Heiko Bozem, Sören Johansson, Horst Fischer, Mira Pöhlker, Bruna Holanda, Ovid Krüger, Christopher Pöhlker, Thomas Klimach, Meinrat Andreae, and Ulrich Pöschl for sharing their data with us. Special thanks are given to the former students Dominique Lörks, Niels Leif Bracher, and Sreedev Sreekumar (formerly all with the Institute for Environmental Physics, University of Heidelberg, Germany) for assisting in the missions.



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
