# Peer review of "Airborne glyoxal measurements in the marine and continental atmosphere: Comparison with TROPOMI observations and EMAC simulations"

_Atmospheric Chemistry and Physics, 2022_

## Author Response (AR1)

We are very grateful to the reviewers for their comments, to which we react in the following way.

The reviewer's comments are written in **bold**, our responses are marked with AC (authors comments).

REVIEW 1:

**Major comments:**

1. **One of the main concerns in the CHOCHO retrieval is the impact of aerosols, especially for those emissions from biomass burning but also over highly polluted regions. Probably not accounting to aerosols could lead to overestimation of glyoxal from airborne and also satellite observation, and thus more discussion in this direction need to be introduced. Have you evaluated the impact of aerosols to the total error in your retrieval? How much is it?**

AC: For the radiative transfer simulations, multiple regionally and seasonally averaged aerosol profiles are used (based on LIVAS and SAGEII observations, see our response to point 31). The impact of the assumed profile on the glyoxal retrieval (e.g. underestimation of the actual aerosol load in cases of strong local emissions) is quantified by additionally employing an aerosol profile representative for highly polluted aerosol conditions in the radiative transfer simulations (Fig. 1, panels c and d). The simulated glyoxal concentrations and VCDs (Fig. 1, panels a and b) based on the assumed aerosol for average (AP1) and severely polluted (AP2) conditions are shown below, exemplarily for a research flight conducted partly over the Upper Silesian Coal Valley (high aerosol load) on May 29, 2018. Evidently, the Nadir retrieved VCDs are only moderately sensitive to the employed aerosol profile, which in the case of occasionally local enhanced aerosol may lead to an average deviation in the inferred glyoxal VCDs of up to 20%. The small sensitivity of our Limb retrieved glyoxal concentrations to the assumed aerosol profile is a result of the employed scaling method, which analyses glyoxal relative to a second (scaling) gas of which the detection is equally affected by aerosols.

[Figure]

Fig.1: Impact of the assumed aerosols profiles in the RT simulations on the glyoxal retrieval (panels a and b), when using an average (panel c, AP1) and an elevated (panel d, AP2) aerosol optical depth profile for the research flight over Germany and the Upper Silesian Coal Valley on 29 May 2018.

2. **Several glyoxal plumes are identify from biomass burning, but also anthropogenic emissions with four plumes corresponding to high amounts of glyoxal, and thus expected from these specific hot-spots. However, under non extreme events, glyoxal detection is low and probably model simulations could correspond more to real glyoxal amounts on average than to those specific scenarios, which could lead to amiss conclusions in this study.**

AC: We are not totally sure about the meaning of the reviewer's comment. We are aware of possible deviations of modelled and measured glyoxal (or for that matter also of air- and spaceborne measurements) when observing locally confined emission events. We argue that such discrepancies are mostly a consequence of the moderate spatial resolution of the model as well as missing details in the emission of glyoxal and it precursors as compared to the airborne glyoxal observations. Consequently, we perform the model-measurement comparison separately for identified emission plumes and the background atmosphere (i.e. outside these plumes) and also distinguish this in Fig. 10 of the manuscript (compare lines 711 and following).

3. **Although, the authors mentioned that altitude criteria selection of flights is done, not clear reason why the difference between TROPOMI and mini DOAS come from, since both perform similar geometric observation and not difference should be expected between satellite and airborne above the altitude selection.**

AC: Please consider our discussion of the different viewing geometries of air- and spaceborne measurements in the replies to the editor's comments. In order to clarify this point, we added section 2.1.4 to the manuscript and additionally also included the figure below (now Fig. 2 in the manuscript).

[Figure]

Fig. 2: Box air mass factors ($B_j$, lower x-axis in panels a and c) of two simulated mini-DOAS Nadir measurements in the lower (panels a and b) and upper (panels c and d) troposphere for different surface albedos between 0.1 and 0.5 (colour-coded). Both simulations are performed for a research flight leading from Oberpfaffenhofen (Germany) to Sal (Cape Verde) on 7 Aug. 2018. The product of

$B_j$ with the assumed a priori glyoxal profile $[X]_j$ (upper x-axis of panels a and c, black line) yields the relative contribution of each altitude layer $j$ (panels b and d). Evidently, airborne Nadir measurements at lower altitudes predominantly, but not exclusively, probe the atmosphere below the flight altitude (panels a and b), whereas measurements in the upper troposphere are sensitive to almost the whole atmospheric glyoxal column density (see the relative fractions in panel b and d).

4. **Finally, why the focus is only CHOCHO but not on formaldehyde (HCHO), which is also possible to retrieve with the mini-DOAS system. In addition, it is well known that this species is also emitted from biogenic, anthropogenic and pyrogenic emissions and behave similar to CHOCHO?**

AC: We agree about the detectability and similarities in the spatial and temporal distribution and fate of atmospheric formaldehyde and glyoxal. In fact, they motivated the simultaneous analysis of glyoxal and formaldehyde in our previous study on carbonyl measurements over the Amazon (Kluge at al., 2020, https://acp.copernicus.org/articles/20/12363/2020/). However, we would like to emphasise that the present study intentionally focuses on the retrieval of glyoxal and on the inferred science. A general overview over the other related carbonyls is beyond the scope of this study, which is already of considerable length. However, an extensive study on coincident retrievals of formaldehyde (and methylglyoxal) and the analysis thereof is the present subject of ongoing research.

**Specific comments:**

5. **P2, Line 25. 'potentially from sea', would you please clarify in which direction the differences occur**

AC: We changed line 24 to: '… but a notable glyoxal overestimation of the model exists for regions with high emissions of glyoxal and glyoxal producing volatile organic compounds (VOC) from the biosphere (e.g. the Amazon). In all other investigated regions the model under-predicts glyoxal to varying degrees, in particular when probing mixed emissions from anthropogenic activities …'

6. **P2, Line 53. Glyoxal lifetime depends on environment conditions like urban or it is similar in all environments?**

AC: In the daytime atmosphere, glyoxal is mainly removed via photolysis and to a lesser degree by oxidation (with OH), or heterogeneous reactions (SOA formation). Generally, with photolysis being the major removal pathway, the lifetime is expected to be similar for a similar illumination in different air masses.

7. **P5, Line 145, what is the effect in the glyoxal retrieval due to the low spectral resolution in the visible range?**

AC: Clearly the spectral resolution reduces the differential optical density as compared to a high resolution glyoxal spectrum (e.g. Volkamer et al., 2005). This is indicated in Fig. 3, where we compare this high resolution spectrum of glyoxal with the differential optical density employing a 0.5 nm (e.g Sinreich et al., 2010) with the 1 nm spectral resolution of the present study. Though notable, the difference in the differential optical density between the 0.5 nm and 1 nm resolution is small (about 10%).

[Figure]

Fig. 3: Effect of different spectral resolutions (panel b) on the differential optical density of glyoxal (panel a).

**8. P6, Table 1 and Table 2, why in the retrieval of glyoxal from mini DOAS system, different cross-section is used than this used for the satellite retrieval, e.g. Pope and Fry (1997) instead of Mason at el. (2016), what is the impact of this cross-section in the validation?**

AC: Since both liquid water absorption cross sections have comparable spectral resolutions (2 nm and 2.5 nm, see Mason et al., (2016) and Pope et al., (1997)) and are only marginally different in the wavelength range of interest, we kept the absorption cross of Pope and Fry (1997) for the present analysis for comparison reasons with the Kluge et al., 2020 study.

**9. P6, Table 2. Why CHOCHO retrievals are in different wavelength? Why not used same spectral range and how glyoxal depends on these changes? Could you quantify it?**

AC: See our responses to point 11.

**10. P7, Line 162. Despite that the authors mention that not significantly effect is observed in the $NO_2$. Is the case for all type of environments? Because previous studies demonstrate that the cross-correlation between NO2 and CHOCHO is significant under anthropogenic emissions.**

AC: We do not state that absorption due to $NO_2$ can generally be neglected in the analysis, but that the addition of a second (colder) $NO_2$ negligibly affects the glyoxal retrieval. However, a (warm) $NO_2$ cross section taken at temperatures representative for the lower troposphere is included in all spectral retrievals. In the discussion of our glyoxal retrieval scenario, we simply point out that our test retrievals indicated an insignificant sensitivity of the inferred glyoxal to the inclusion of an additional colder (!) $NO_2$ cross section in the spectral retrievals. Fig. 4 shows two example glyoxal retrievals (excluding and including an additional colder $NO_2$ cross section) for a Nadir measurement (which should be more sensitive to boundary layer $NO_2$ than the Limb measurement due to the smaller AMFs/shorter light paths) under elevated $NO_2$ (simultaneously Nadir detected by mini-DOAS) over the Munich metropolitan area on 7 Aug. 2018. Evidently, the additional cold $NO_2$ absorption cross section (at 223K) has insignificant impact on the fit quality, the residual structure and the inferred glyoxal.

[Figure]

Fig. 4: Glyoxal retrieval in Nadir direction for elevated near surface NO2, excluding (left column) and including an additional colder NO2 cross section (at 223K) in the analysis for a measurement over the Munich metropolitan area on 7 Aug. 2018.

**11. P7, Line 170. As the authors pointed out that the fitting window has been changed to a continue one in comparison to previous study by Kluge et at. (2020) yield to improvements in the spectral residuum and signal to noise ratio, how much large are these improvements (e.g. residual and noise)? What is the relative difference in CHOCHO SCD between previous retrieval and the one used in this study?**

AC: In the analysis, the impact of the wavelength range of the glyoxal retrieval is investigated by applying both suggested fitting windows to all measurements. Fig. 5 indicates that relative to the two discrete fitting windows, the continuous wavelength range decreases the spectral residual on average by a factor of three, for comparable signal to noise ratios. For measurements in the free troposphere, the resulting dSCDs based on the continuous spectral analysis are systematically larger than those obtained based on the discrete fitting window (Fig. 5, panels c and f). This is mainly caused by the larger scatter of the dSCDs around zero when using the discrete wavelength range.

[Figure]

Fig. 5: DOAS retrieval of glyoxal from all 72 research flights in the Limb (upper row) and Nadir (lower row) viewing geometry based on two different spectral ranges avoiding (FS2gap, red) and including (FS1cont, green) the 7$v$ absorption band of water, respectively, by using two fitting windows ranging from 430 or 435 nm to 439 nm and 447 to 460 nm, or a continuous spectral range from 430 or 435 to 460 nm, respectively. The data are averaged on a 1 km altitude grid. The different panels show the spectral residuum (a, d), the signal to noise ratio (S/N) of the retrievals (b, e) and the resulting dSCDs (c, f).

**12. P7, Line 178. What different ambient conditions? Is the retrieval of CHOCHO depending on the ambient conditions? If it is the case, why the CHOCHO retrieval should dependent the source producing it?**

AC: Changing ambient conditions (i.e. due to colder/warmer ambient temperatures) do not impact the measurements and spectral retrieval per se, but were considered in the choice of the absorption cross section temperatures employed in the spectral retrieval.

**13. P7, Line 179. For off axis DOAS observations the impact of water vapour is significant and cross correlation is more evident for weak absorber such as glyoxal. However, in nadir observation the effect of water vapour in the glyoxal retrieval would be expected rather smaller. Can you quantify this effect in your retrieval and also have you evaluate the dependency of glyoxal retrieval on the water vapour cross-section used?**

AC: Since we report on airborne Nadir and Limb measurements, the line of sight column densities of $H_2O$ are generally much smaller than in off-axis measurements from the ground. Accordingly, retrieval exercises using different $H_2O$ absorption cross sections indicated little dependence on the inferred glyoxal in accordance with the exercises when including and excluding the $7\nu$ absorption band of water (see our responses to point 11).

**14. P7, Line 188. Would you please write to which location and altitude the reference spectrum corresponds for the different flights?**

AC: As explained on page 7, lines 187 to 190, the reference Fraunhofer spectra are separately selected for each flight. This choice compromises between a) minimal solar zenith angle, b) minimal cloud fraction, and c) maximal flight altitude for the sake of minimizing the amount of glyoxal. Differently to satellite measurements, we use in-flight references for our airborne retrieval, such that the present data set contains 72 different reference spectra, all taken at different locations and altitudes (however always from measurements in the upper troposphere). While the exact locations of each reference can certainly be provided upon request, we fear a table containing all 72 flights would distract from the scientific scope of the paper rather than help the reader.

**15. P9, Line 240, Would you please clarify why VCD does not correspond to the total vertical column and only a fraction?**

AC: See the answer to point 3.

**16. P10, Line 243. Would you please give a reference or clarification why the 8 km threshold is used for selection of for flight in the comparison?**

AC: See the answer to point 3.

**17. P13, Line 338. How large is the spectral variability of mini DOAS system regarding temperature changes in the spectrometer?**

AC: An extensive discussion of the mini-DOAS temperature stability and the temperature dependence of the point slit function can be found in Hünecke (2016) (for download of the PhD thesis see here https://archiv.ub.uni-heidelberg.de/volltextserver/22573/). For low temperatures, the point spread function of the instrument was found to vary by 0.005nm/K. The temperature changes were usually below 1K for flights predominantly performed in the upper troposphere up to several K towards the end of long duration flights within the boundary layer (i.e. at large ambient Temperatures). According to our test measurements in the laboratory, we therefore discarded measurement above a temperature change large than 3K.

**18. P13, Line 348. Would you please describe how is applied the empirical correction in case of extreme NO$_2$ absorption?**

AC: In order to clarify, we added after line 387: 'As described in detail in Lerot et al., (2021), this correction is based on a linear regression fit obtained by a representative sensitivity test for glyoxal measurements at NO2 SCDs larger than $2 \times 10^{16}$ molec cm$^2$.'

**19. P14, Line 354. Are aerosols accounted in the (satellite) retrieval?**

AC: See the answers to point 31.

**20. P17, Figure 3, The plumes 1.1, 2, etc., are not scaled? The text is a bit confuse, please make clear it.**

AC: The text states: 'All four plumes are not shown in full scale for better comparability with the other profiles.' Instead, for all (truncated) plumes, the maximum glyoxal mixing ratios are given in the legend.

**21. P20, Line 495. Is the plume 1.2 also observed from TROPOMI? If yes, how this plume compare to these from Mini DOAS system? How was the evolution of the plume since the fire started? What is the age of the plume?**

AC: The correlation of the two instruments within plume 1.2 is described in detail by Fig. 8, panels c and d. Unfortunately, the latter two questions can't be answered for observations from fast moving instruments deployed on aircrafts or satellites, but only from stationary i.e. ground-based instruments.

**22. P21, Line 524. Why glyoxal from Mediterranean Sea are larger than from Amazon rainforest? Normally, glyoxal from biogenic emissions are expected higher than anthropogenic or any other source with exception from biomass burning. Does TROPOMI observe similar behaviour between Mediterranean region and Amazon rainforest?**

AC: As stated in the manuscript, such findings are related to the per-se biased aircraft observations, which are often preferred to probe in particular polluted air masses (e.g. in biomass burning plumes, pollution from larger cities at the coast of the Mediterranean Sea, et cetera rather than background air). On average, biogenic emissions are globally the most important glyoxal source. However, unsurprisingly, local emission plumes (anthropogenic or biomass burning) cause glyoxal enhancements which exceed its biogenic background. Correspondingly, over the Amazon rainforest, the largest glyoxal was observed in biomass burning plumes and not in the general rainforest background (compare Kluge at al., 2020). Equally, the probing of fresh anthropogenic and biomass burning plumes near the Mediterranean coasts yields larger glyoxal than observations in its average biogenic background. Obviously, this pattern is not expected when probing air over the remote Mediterranean or when avoiding local emission plumes. Due to this observation related bias in aircraft studies and since the sampling of TROPOMI (a) is not aimed to selectively probe pollution plumes and (b) is significantly less sensitive to locally confined low lying pollution plumes (as compared to the airborne measurements), the glyoxal measurements from both instruments in this region can't and should not be compared in a climatological sense but only along the specific flight trajectories.

**23. P22, Line 540. Previous studies usually shown low glyoxal values over Europe and the Mediterranean Sea (Lerot et al. 2021, Alvarado, et al 2014, Chan Miller et al 2014), however in this study enhanced values are found over this region. How can be it explained? Could be related to the spatial resolution of satellite observations that low values are observed from Satellite for these regions?**

AC: The cited studies are all from satellite observations, which are better suited to probe 'climatological' fields of pollutants than aircraft. However as stated in the manuscript, the aircraft observations were curtailed (and thus biased) to probe pollution plumes in particular. In this respect, the comment and the questions more point to the problem of comparing the same things. Please inspect again our responses to point 22.

**24. P23, Line 594. Why do not use all the flights? Glyoxal is expected be found close to the surface for biogenic and anthropogenic emissions. To what altitude correspond those considered low flying aircraft? Do you expect glyoxal at 7 or 8 km altitude?**

AC: See the answers to point 3.

**25. P24, Line 609. Although, to use large grid reduce the noise level in the satellite detection, a smaller grid could lead to more accurate comparison between mini DOAS system and TROPOMI. How the comparison looks using the smallest pixel size of TROPOMI versus mini DOAS system for large glyoxal plumes?**

AC: As stated in the manuscript, individual TROPOMI measurements of glyoxal at low pollution are not very meaningful due to the large noise (for details see the answer to point 29).

**26. P24, Line 611. What is the delta time between observations (TROPOMI versus mini DOAS)? There is any criteria for it?**

AC: Since the mini-DOAS flight time varies daily and the TROPOMI overpass time depends on the latitude (13:30 LT equator crossing time), the delta time varies for the individual research flights. During some measurements, both overpasses occurred within minutes, while for others the measurements are some hours apart (≤ 4h due to the available daylight and the typical timing of individual flights). Consequently, we do not expect both sets of observations to strictly correlate when observing locally confined emission events and for such comparisons the measurement times are indicated in the legend. (e.g. Fig. 8 of the manuscript).

**27. P24, Line 619. $3.2x10^{14}$ variability correspond to the standard deviation?**

AC: All uncertainty ranges are given as median absolute deviation.

**28. P27, Figure 8a. Please make more visible the legend for TROPOMI and mini-DOAS or move outside of figure.**

AC: Fig. 8, panel a has been changed accordingly.

**29. P28, Figure 9. Although the good consistency between both data sets, the TROPOMI present more negative values than mini-DOAS. Do you know why is the case?**

AC: For individual measurements the TROPOMI retrievals are considerably more noise affected than those made from the aircraft (by a factor of ~6, for details see our response the comments of the editor). In consequence, the satellite measurements are more strongly affected by statistical scatter than the aircraft measurements, which for very low glyoxal leads to relatively more negative results.

**30. P29, Line 685. Has the missing of additional NO2 cross-section impact in the mismatching between TROPOMI and mini-DOAS over these polluted region?**

AC: See the responses to point 10 and Fig. 4.

**31. P29, Line 687. How is correct for aerosols in the TROPOMI and mini-DOAS retrievals?**

The McArtim radiative transfer model used for the mini-DOAS retrieval accounts for aerosols by including an average, regionally characteristic aerosol profile obtained by LIVAS (Amiridis et al., 2015) and the Stratospheric Aerosol and Gas Experiment II (SAGE II). The dependency of the retrieval on this profile is described in point 1 and Fig. 1. The TROPOMI retrieval discards observations affected by dense aerosols and clouds by applying a cloud filter (effective cloud fraction below 20%, compare Lerot et al., 2021). In agreement to the mini-DOAS aerosol sensitivity discussed in question 1, this

may lead to an increased uncertainty of the air mass factor in the order of 15% for measurements e.g. affected by strong biomass burning plumes (Lorente et al., 2017).

**32. P30, Line 731. Despite the large underestimation from the model, the variability of glyoxal from mini-DOAS is large, which also could lead to miss interpretation of the figure. For those days where the glyoxal variability is large. To what is it associated? Seems to be that TROPOMI (figure 9 vs figure 10) observations match better to the model than the mini-DOAS. How is it explained?**

AC: The larger variability in glyoxal for the airborne measurements is most likely due to glyoxal emissions and glyoxal secondary formation, both of which are not captured in detail by the model. Likewise, as stated in the manuscript, the satellite has difficulties to detect glyoxal plumes of limited vertical extent (and hence with a relatively small contribution to the total atmospheric VCD) as well as to detect faint glyoxal plumes over surfaces of low albedo. It is not very astonishing that the aircraft may provide a more detailed and comprehensive (spatially but not temporally resolved) picture of glyoxal in the atmosphere than the satellite or model are able to provide.

**33. P34, Line 817-825. Despite that different plumes are observed and associate to specific events some of them present very high glyoxal amounts, however not for all fire events. Are those plumes observed similar spatially distributed? The amount of glyoxal depends on the precursor emitted from the fire or the type of vegetation? The altitude at which the plumes from fire are injected in the atmosphere play a role in the interpretation from fire emissions?**

AC: In part, we concur with the statements of the reviewer (see our responses to point 32). However, the reviewer and the reader should consider that the present airborne measurements were not specifically designed to monitor glyoxal in different biomass burning plumes (contrary to those for example presented in Kluge et al., 2020) but rather were made within the frameworks and objectives of the different research missions. In this respect, the questions of the reviewer need to be tackled by more dedicated missions in the future.

REVIEW 2:

1. **Abstract, line 16: remove "over days"**

AC: We changed line 16 accordingly to: 'Our observations of glyoxal in such aged biomass burning plumes, ...'.

2. **Abstract, line 28: suggest "mixture" rather than "cocktail", which is colloquial**

AC: We changed line 28 accordingly to: '...from the degradation of the mixture of...'.

3. **Lines 60-62: Please add the Washenfelder et al. reference for measurements in Los Angeles. Washenfelder, R.A., et al., The glyoxal budget and its contribution to organic aerosol for Los Angeles, California during CalNex 2010. J. Geophys. Res., 2011. 116: p. D00V02.**

AC: We added the suggested publication.

4. **Line 133: "or briefly" appears to be out of place and likely not intended**

AC: We changed line 133 to: '...the collisional complex $O_2$-$O_2$ (further on called $O_4$; ...)'.

5.  **Line 191 and following paragraph: Specify the integration time associated with the LOD for the glyoxal dSCD or mixing ratio.**

AC: Since the statistical noise is given by the number of collected photons ($N_{PH}$), for sufficiently large $N_{PH}$ the measurement noise should be dominated by the read-out noise. At the same time, increasing $N_{PH}$ also increases the spectral integration time and hence enlarges the spatial resolution of the measurement along the flight track of the (moving) aircraft. Measurements prior to the field deployment of the instrument indicate that an optimal compromise of the photoelectron shot noise, read-out noise, and resolution is obtained when accumulating 100 read-outs at 60% saturation and 300ms exposure time each. In order to clarify, we added in line 191: 'Based on an exposure time of 300ms, a saturation of 60%, and 100 added readouts (30s integration time), the mini-DOAS detection limit for glyoxal...'.

6.  **Line 204: Remove extra parentheses**

AC: We corrected line 204 accordingly to: '…correction factors $\alpha_{x,j}$ and $\alpha_{O4,j}$ to quantify…'.

7.  **Line 320: Should biogenic influenced air be on this list? Is this part of pristine continental?**

AC: We are not totally sure about the reviewer's comment, since the paragraph following line 318 exclusively addresses air masses predominately affected by anthropogenic pollution.

8.  **Line 348: Is the nature of the empirical correction of CHOCHO at high NO2 explained in Lerot 2021? Is it possible to summarize briefly here?**

AC: The empirical correction of the TROPOMI glyoxal retrieval for strong $NO_2$ is described in detail in Lerot et al., (2021), section 3.1.2. In order to clarify, we added after line 387: 'As described in detail in Lerot et al., (2021), this correction is based on a linear regression fit obtained by a representative sensitivity test for glyoxal measurements at $NO_2 > 2x10^{16}$ molec $cm^2$.'

9.  **Line 394: Why are direct emissions of glyoxal from biomass burning excluded?**

AC: For the comparison, an EMAC set-up has been used that represents recent EMAC studies focusing on up-to-date VOC photochemistry. Direct biomass burning emissions of glyoxal were not included in these set-ups so far. To be in line with those studies, we also did not include these biomass burning emissions in the present comparison. An optimised EMAC set-up, which also includes direct biomass burning emissions of glyoxal, is the focus of our companion paper by S. Rosanka, which is currently under preparation.

10. **Table 3: Values are listed as median with standard deviation – should this read mean and standard deviation? Is there a reason for using a median rather than a mean in combination with a standard deviation. Would percentiles be more appropriate, and if so, is the glyoxal in each region normally distributed?**

AC: We replaced the standard deviation by the median absolute deviation throughout the manuscript in order to use an internally consistent statistical measure.

11. **Line 488: The large emission of isoprene is associated with the data from South East Asia but not the Amazon basin?**

AC: Yes. For clarification, we changed line 529 to:'... than reported from the tropical rainforest in a rural region of South-East Asia (up to 1.6 ppb) likely due to large emissions of isoprene there…'.

12. **Line 562-564: Min et al. also report airborne glyoxal over the Eastern U.S. that can be listed here. Min, K.E., et al, A broadband cavity enhanced absorption spectrometer for aircraft**

**measurements of glyoxal, methylglyoxal, nitrous acid, nitrogen dioxide, and water vapor. Atmos. Meas. Tech., 2016. 9(2): p. 423-440.**

AC: We added the suggested publication.

13. **Figure 8: Figure is rather difficult to read and would benefit from an overview map identifying the locations of each of the different examples. The insets in the individual figures do not have sufficient reference to understand where each images it taken easily. The labels, (a), (c), (e) and (f) are quite difficult to see.**

AC: We agree and adjusted the insets in Fig. 8, panels a, c, f in order to provide a better reference to the geographic location of each measurement and shifted the panel labels for better visibility.

14. **Line 686-688: Measurement noise that leads to negative glyoxal values implies that there is also noise that leads to larger positive values than are actually present in the atmosphere. Inclusion of the positive noise with omission of the negative noise then biases the model – measurement subtraction in Figure 10. Does the one-sided omission of noise bias this comparison? If so, by how much? If not, the authors should justify.**

AC: Each single measurement is affected by a statistical uncertainty, such that $n$ repetitions of the measurement under equal conditions would lead to a normal distribution around the mean value $x$. Due to the relatively larger uncertainty of the measurements of small as compared to those of large glyoxal concentrations, the relative width of this distribution is expected to be larger for small glyoxal than for large glyoxal observations. For low glyoxal, the respective scatter around the mean may therefore cause negative values in the retrievals. While such negative retrievals are clearly caused by statistics, for large glyoxal there is no robust method to differentiate between statistically overestimated glyoxal (relative to the mean value) and the actual glyoxal. Therefore, in comparison to model simulations which do not produce negative values, for statistical consistency in the two sets of samples the negative retrievals are discarded (Fig. 6, black) rather than to keep them in the analysis (Fig. 6, red). The comparison of both approaches however indicates that including/excluding negative measurements causes only minor changes to the results, which are all within the uncertainty of the given median differences of the observations and the model (Fig. 7).

[Figure]

Fig. 6: Median difference of observed and simulated glyoxal in the different regions including negative observations (red) and excluding negative observations (black).

[Figure]

Fig. 7: Difference of the above plotted (Fig. 1) model to measurement glyoxal comparison including ($G_{obs,noNeg.}-G_{sim}$) and excluding negative observations ($G_{obs,all}-G_{sim}$).

**15. Line 705: Should this read overestimated (not underestimated) emissions of long lived precursors?**

AC: We agree, that the much larger simulated than observed glyoxal in the boundary layer and the free troposphere indicates overestimated rather than underestimated concentrations of longer-lived glyoxal precursors. We consequently changed 'underestimated' to 'overestimated' in line 735.

**16. Line 800: Glyoxal may also be a product of multi-generation biomass burning oxidation rather than simply long-lived precursors.**

AC: For clarification, we adjusted line 833-835 to: '..., the glyoxal detected in these aged biomass burning plumes was necessarily secondarily formed from direct or multi-generation oxidation of yet unidentified longer-lived VOC precursor species (e.g. benzene, acetylene, or aromatics), which were co-emitted during the wildfires.'

**17. Line 805-809: The biomass burning source is a potential explanation for the tropical oceanic glyoxal, but it would have a very different vertical distribution than a surface glyoxal source. To what extent are the vertically resolved data from this analysis consistent with either source?**

AC: To clarify, we added after line 812: 'This finding is supported by the inferred vertical glyoxal profiles over the different marine regions. When comparing the vertical glyoxal profiles above the tropical Atlantic to those over the mid- and high-latitude Atlantic, the relative enhancement of glyoxal in the tropics appears restricted to the tropical marine boundary layer. At higher altitudes, the glyoxal profiles over different regions of the Atlantic are similar (see table 3). This finding strongly points to a marine glyoxal source in the tropics rather than long-range transport of glyoxal and its precursors from terrestrial emissions. If the latter process was the dominant glyoxal precursor in the observed marine air masses, elevated glyoxal would also be expected at higher altitudes and latitudes, and not exclusively in the tropical boundary layer.'

PUBLIC COMMENT:

Dear Mriganka Sekhar Biswas,

thank you very much for your comments (indicated in **bold**) regarding our manuscripts. Detailed replies to your questions are outlined below.

Please note however that we do not report on MAX-DOAS but rather on airborne Limb (at constant elevation angle) and Nadir measurements using DOAS in the spectral retrieval.

**1. $NO_2$ retrievals using different $NO_2$ reference spectra**

Since high $NO_2$ was predominantly encountered when flying within the planetary boundary layer, our glyoxal retrieval accounts for $NO_2$ absorption by including a $NO_2$ absorption cross section at 294K (compare lines 165 and following as well as tables 1 and 2 of the manuscript). For more details on the sensitivity of the inferred glyoxal as a function of $NO_2$, please inspect our response to comment 10 of Anonymous Referee #1. There, for large $NO_2$ conditions two example retrievals of glyoxal are compared, including and excluding a second $NO_2$ cross section.

For your orientation how well our $NO_2$ measurements compare with others, simultaneously measured $NO_2$ was investigated in a comparison study including several $NO_2$ instruments deployed on two aircrafts as well as photochemical modelling, see Schumann et al., https://zenodo.org/record/4427965 , fig. 49).

**2. Glyoxal retrievals using different wavelength ranges**

In the analysis, the impact of the wavelength range for the glyoxal retrieval is investigated by applying both suggested fitting windows to all measurements. The test retrievals indicate that relative to the two discrete fitting windows, the continuous wavelength range decreases the spectral residual on average by a factor of three for comparable signal to noise ratios. For more details, please inspect our responses to comment number 11 of the Anonymous Referee #1. It also shows a figure on the impact of the employed wavelength range on the inferred dSCDs, signal to noise, and spectral residuum.

**3. Figures 6 and 7**

In fact, the idea of Figs. 6 and 7 is to show where the distributions coincide and in which cases they do not. For better comparability, both distributions are intentionally plotted one top of each other, such that for an ideal comparison one would only see a single distribution.

With my best regards,

Flora Kluge

Bibliography:

Alvarado, L. M. A., et al. "An improved glyoxal retrieval from OMI measurements." *Atmospheric Measurement Techniques* 7.12 (2014): 4133-4150.

Amiridis, Vassilis, et al. "LIVAS: a 3-D multi-wavelength aerosol/cloud database based on CALIPSO and EARLINET." *Atmospheric Chemistry and Physics* 15.13 (2015): 7127-7153

Chan Miller, C., et al. "Glyoxal retrieval from the ozone monitoring instrument." *Atmospheric Measurement Techniques* 7.11 (2014): 3891-3907.

Kluge, Flora, et al. "Profiling of formaldehyde, glyoxal, methylglyoxal, and CO over the Amazon: Normalised excess mixing ratios and related emission factors in biomass burning plumes." (2020).

Lerot, Christophe, et al. "Glyoxal tropospheric column retrievals from TROPOMI–multi-satellite intercomparison and ground-based validation." *Atmospheric Measurement Techniques* 14.12 (2021): 7775-7807.

Lorente, Alba, et al. "Structural uncertainty in air mass factor calculation for NO 2 and HCHO satellite retrievals." *Atmospheric Measurement Techniques* 10.3 (2017): 759-782.

Mason, John D., Michael T. Cone, and Edward S. Fry. "Ultraviolet (250–550 nm) absorption spectrum of pure water." *Applied optics* 55.25 (2016): 7163-7172.

Pope, Robin M., and Edward S. Fry. "Absorption spectrum (380–700 nm) of pure water. II. Integrating cavity measurements." *Applied optics* 36.33 (1997): 8710-8723.

Sinreich, R., et al. "Ship-based detection of glyoxal over the remote tropical Pacific Ocean." *Atmospheric Chemistry and Physics* 10.23 (2010): 11359-11371.

Volkamer, Rainer, et al. "High-resolution absorption cross-section of glyoxal in the UV–vis and IR spectral ranges." *Journal of Photochemistry and Photobiology A: Chemistry* 172.1 (2005): 35-46.

---

## Author Response (AR2)

Dear Andreas Richter,

please find below our reactions and responses (in bold) to your comments and suggestions (in italic) for the manuscript acp-2022-416, dated from Dec. 12, 2022.

*Dear Flora Kluge,*

*I'm pleased to accept your revised manuscript "Airborne glyoxal measurements in the marine and continental atmosphere: Comparison with TROPOMI observations and EMAC simulations" for publication in ACP subject to minor revisions as listed below.*

*Most of the concerns raised by the reviewers have been addressed in the replies, and some clear improvements have been implemented in the revised manuscript. Please consider the following minor before submitting the final version of your manuscript:*

**Thank you for the overall positive assessment of our reactions to the previous reviews of the manuscript.**

*• In several places of the manuscript, you mention that the mini-DOAS measurements are validated by the satellite observations. I don't think that the satellite observations, in particular at low glyoxal levels can and should be used to validate your observations. If anything, it should be the other way round.*

**We agree and accordingly changed the wording to 'cross validation' of the air- and spaceborne measurements, wherever necessary in the manuscript** (lines 107, 113, and 609)**.**

*• On page 5, 153 you state "the instrument measures the atmospheric column density of the targeted gases below the aircraft..". I do not think that is correct (see your Figure 2). I suggest to reformulate to "the instrument receives light from the surface and atmosphere below the aircraft ..:" or a similar formulation.*

**We accordingly changed the text to '…In the Nadir observation mode, the instrument receives light from the surface and atmosphere below the flight altitude (Fig. 2). It thus preferably measures glyoxal below the aircraft with a rectangular foot print of …'**

*• On page 8, first lines you provide the detection limit for the column. As this is still in the slant column section, I assume that you are providing slant column detection limits, and these are independent of the air mass factor.*

**We concur and accordingly changed the text to '…the typical SCD detection limit is….' in order to make more clear what is meant.**

*• In the caption of Figure 2, please add the information on what you assumed in terms of clouds (I assume none) and aerosols (I don't know what you assumed here).*

**In order to make clear what is assumed in the radiative transfer simulations, we added the following sentence to the legend of Figure 2:  For the radiative transfer simulations, no clouds, but aerosol profiles as described in the text are assumed (see sect. 2.1.4).**

*• On page 12, line 290 you state that you have used an "all-sky albedo of 0.3". Is that really the case? This value appears excessively large to me. If this is meant to correct for the effect of residual clouds,*

*this would depend strongly on the glyoxal vertical profile and the cloud altitude – assuming a large albedo increases the AMF while a cloud above the boundary layer should lead to a smaller AMF.*

**Essentially we following your comment, but fortunately the sensitivity of the AMFs on the assumed all sky albedo for the actual flight altitudes is relatively moderate but distinctively larger for altitudes below the aircraft, as the RT simulations in Figure 2 indicate. In order substantiate our conclusions regarding the RT related uncertainties on the inferred VCDs, we additionally compare the inferred VCDs for clear (A = 0.1), cloudy (A = 0.6) and all skies (A = 0.3) for the glyoxal profiles and flight altitudes shown in Figure 2 (and mention them in the text at the end of section 2.1).**

*• On page 31, line 717 you state that negative columns are omitted in the comparison to the model data. As the second reviewer, I believe that this will introduce a bias in your comparisons. This bias may be small, but removing negative values in such a comparison is not mathematically correct. As you have already evaluated the effect in your reply to the reviewer, I would suggest to replace the figures and numbers with the version using all measurement data.*

**We agree and adjusted Figure 12 as well as the text accordingly (i.e. removal of lines 716ff: 'Since for low glyoxal concentrations the measurement noise occasionally leads to negative data (see Fig. 6 and Fig. 11), and the model does not reproduce such measurement noise, inferred negative glyoxal observations are omitted from the comparison.').**

*• In the conclusions, please repeat the point that your profiles should not be used as a climatology as the sampling is strongly biased to polluted and coastal scenes. While this is clearly stated in the text, I worry that some readers will directly skip to using the profiles as "typical" for the respective regions, which would not be good.*

**We accordingly changed the sentence in line 855 (cont.) to:**
**'The combined airborne Nadir and Limb data set is the first of its kind and may offer new information into the fate of atmospheric glyoxal in the global atmosphere. The Limb measurements further allow us to infer and investigate different glyoxal profiles in the troposphere over different regions of the world. They are, however, not representative for climatological glyoxal studies due to their limited coverage in space and time.'**